



# Extreme wet seasons – their definition and relationship with synoptic scale weather systems

Emmanouil Flaounas[1], Matthias Röthlisberger[1], Maxi Boettcher[1], Michael Sprenger[1] and Heini Wernli[1]

[1]Institute for Atmospheric and Climate Science, ETH Zürich, Zürich, Switzerland

*Correspondence to*: Emmanouil Flaounas (emmanouil.flaounas@env.ethz.ch)

**Abstract.**

An extreme aggregation of precipitation on the seasonal timescale, leading to a so-called extreme wet season, can have substantial environmental and socio-economic impacts. In contrast to extreme precipitation events on hourly to daily timescales, which are typically caused by single weather systems, an extreme wet season may be attributed to a combination
of different and/or recurring weather systems. In fact, extreme wet seasons may be formed by almost continuously occurring moderate events, or by more frequent and/or more intense short-duration extreme events, or by a combination of these scenarios. This study aims at identifying and statistically characterizing extreme wet seasons around the globe, and elucidating their relationship with specific weather systems.

To define extreme wet seasons, we used 40 years (1979-2018) of ERA-Interim reanalyses. Primary extreme seasons were defined independently at every grid point as the consecutive 90-day period with the highest accumulated precipitation. Secondary extreme seasons were also considered, if accumulated precipitation amounts to at least 90% of the precipitation in the primary season at the same grid point. A high number of secondary extreme seasons was found for instance in the extratropical storm tracks, suggesting that these regions are less likely to experience an exceptional amount of precipitation
in a particular 90-day period. In most continental regions, the extreme seasons occur during the warm months of the year, especially in the mid-latitudes. Nevertheless, colder periods might be also relevant to extreme seasons within the same continent, especially in coastal areas. All identified extreme seasons were statistically characterised in terms of anomalies compared to the climatology of the number of wet days and daily extreme events. Results show that daily extremes are decisive for the occurrence of extreme wet seasons in regions of frequent precipitation, e.g. in the tropics. In contrast, e.g., in
arid regions where wet days are scarce, extreme seasons may occur only due to anomalously high numbers of wet days. In the subtropics and more precisely within the transitional zones between arid areas and regions of frequent precipitation, both an anomalously high occurrence of daily extremes and wet days are related to the formation of extreme wet seasons.

The spatial extent of regions affected by the same extreme wet season is variable and can reach continental scales, although
the vast majority of extreme seasons is limited to scales of the order of $20 \times 10^5$ km². Finally, the relationship of extreme seasons to synoptic-scale weather systems was investigated on the basis of four objectively identified weather systems that



are known to be associated with intense precipitation: cyclones, warm conveyor belts, tropical moisture exports and breaking Rossby waves. A grid-to-grid association of these weather systems to daily precipitation allows quantifying their role for extreme wet seasons. In particular, cyclones and warm conveyor belts contribute strongly to extreme wet seasons in most

regions of the globe. But interlatitudinal influences are also shown to be important: tropical moisture exports, i.e., the poleward transport of tropical moisture, can contribute to extreme wet seasons in the mid-latitudes, while breaking Rossby waves, i.e., the equatorward intrusion of stratospheric air, may decisively contribute to the formation of extreme wet seasons in the tropics. Four illustrative examples provide insight into the synergetic effects of the four identified weather systems on the formation of extreme wet seasons in the Arctic, the mid-latitudes, Australia, and the tropics.

**1 Introduction**

This study focuses on extreme precipitation, however not on short timescales of single weather systems like thunderstorms or cyclones, but on the seasonal timescale. The analysis of extreme precipitation events on timescales of hours to a few days has always been a centrepiece of weather and climate research due to their relevance for a variety of socio-economic sectors. Indeed, many studies investigated single extreme precipitation events to identify the key dynamical and physical processes

involved (e.g., Doswell et al., 1998; Massacand et al., 1998; Delrieu et al., 2005; Holloway et al., 2012; Moore et al., 2012; Winschall et al., 2012; Flaounas et al., 2016). In addition, a high number of climatological studies quantified the relationship of extreme precipitation events with specific synoptic-scale flow systems like cyclones (Pfahl and Wernli, 2012), fronts (Catto and Pfahl, 2013), and warm conveyor belts (Pfahl et al., 2014). Finally, another important strand of research addressed the future evolution of extreme precipitation events in a changing climate, using a plethora of simulation ensembles,

reanalysis datasets and observations (e.g., Easterling et al., 2000; Shongwe et al., 2011; Pfahl et al., 2017). However, environmental risks are not limited to the occurrence of single, outstanding extreme precipitation events, but they are also potentially related to precipitation on longer timescales. For instance, the costliest, hyperactive North Atlantic hurricane season of 2017 had a significant impact on the coastal population of the US (Halverson, 2018) although it did not include any record-breaking intense hurricane. The main impact was due to an anomalous sequence of landfalling tropical cyclones.

Another example of seasonal-scale environmental risk is the direct relationship between the seasonal rainfall over the Sahel and the epidemics of meningitis. An anomalously wet African monsoon season may have a detrimental impact on public health on continental scales (Sultan et al., 2005). The factors contributing to the formation of extreme seasons may not be linked directly to the anomalous occurrence of extreme events as intuitively expected. In fact, Röthlisberger et al. (2020) showed that an extreme hot or cold season may not be always provoked by the repetitive occurrence of exceptionally high or

low temperatures, respectively. In contrast, an extremely warm summer may also be provoked due to its coldest days being anomalously mild. Therefore, the seasonal distribution of weather variables plays an important role to the characterisation of a season. Despite its high socio-economic relevance, the analysis of extreme precipitation seasons has not gained high visibility in climate research so far. This study addresses this research gap and aims to contribute to a better understanding of





the characteristics of extreme precipitation seasons around the globe and to provide insight about the responsible weather

systems. In the following we refer to these seasons as extreme wet seasons.

The definition and identification of distinct precipitation seasons is a delicate issue and highly dependent on the region of interest. Monsoon-affected regions typically experience a clear onset date that signalizes the beginning of the precipitation period (Bombardi et al., 2017; 2019), while several mid-latitude areas experience more than one rainy season or they are

characterized by wet conditions year-round. On the other hand, semi-arid and arid areas do not have clearly preferred precipitation periods due to the scarcity of wet days and thus the definition of a precipitation season becomes less meaningful in these areas (Wu et al., 2007). Regardless, if a region experiences an extreme seasonal aggregation of precipitation, e.g., due to an anomalous frequency of daily extreme events, this has a potentially hazardous effect. Indeed, the above examples about hurricane and monsoon seasons illustrate that significant seasonal precipitation anomalies may be

related to both an anomalous frequency and intensity of precipitation events. Depending on the region, seasonal precipitation extremes may be related to a well-defined, unique, and recurrent weather system, such as tropical cyclones, or they may be related to a variety of weather systems that occur sequentially in the considered season, favoured by regional or global-scale atmospheric conditions. For instance, Davies (2015) and Röthlisberger et al. (2019) showed that the anomalously wet and stormy European winter of 2013/2014 was related to recurrent upper-tropospheric flow conditions that triggered a succession

of high-impact weather systems. Climatological influences might be also important for seasonal precipitation, for instance in the Mediterranean. The grand majority of precipitation in this region is due to intense cyclones (Flaounas et al., 2018), however, the intensity of cyclones and related rainfall is influenced by the North Atlantic Oscillation and the El Nino Southern Oscillation (Mariotti et al., 2002; Raible, 2007). Seasonal precipitation has been the theme of numerous studies in the past; however, in this study we add to the state-of-art by focusing on extreme wet seasons and performing a systematic

analysis of how individual weather system contribute to their occurrence.

Weather systems on different spatial scales may interact to provoke extreme wet seasons. For instance, synoptic-scale atmospheric conditions may favour the occurrence or intensify mesoscale weather systems, which in turn may lead to variable amounts of precipitation depending on their physical characteristics, e.g. water vapour content, precipitation

efficiency, etc. It is a scientific challenge to delineate and objectively identify all links in this chain of events in climatological datasets. As mentioned above, several studies have quantified the role of specific weather systems such as cyclones, fronts, warm conveyor belts, tropical moisture exports, troughs, cut-off systems, and breaking Rossby wave for precipitation on regional and global scales (Knippertz and Wernli, 2010; Catto et al., 2012; Hawcroft et al., 2012; Pfahl et al., 2014; Flaounas et al., 2016). Nevertheless, it is an open question whether these weather systems occur successively or act

synergistically to form an extreme wet season in a certain region. Moreover, it is an open question whether extreme wet seasons may be produced by more frequent daily extreme events, more intense daily extremes, or by higher persistence of moderate rainfall – or a combination of these options. In fact, the aggregated contribution of a weather system to seasonal



precipitation may be statistically characterized by its frequency and its intensity (e.g. Toreti et al., 2010; Moon et al., 2019). This study uses these concepts to statistically characterize extreme wet seasons, to address their spatial coherence, and to

quantify the contributions of specific weather systems. In this way, we aim to provide novel insight into the relationship between the statistical characteristics of extreme wet seasons and their dynamical origin.

In the next section, we present the datasets and methods used to define extreme wet seasons and to objectively identify the contributing weather systems. In section 3, we perform a statistical approach at every grid point to characterize extreme wet

seasons by the number of daily extreme precipitation events and by the number of wet days that occur in this season. The spatial coherence of extreme wet seasons is then analysed in section 4. Section 5 shows examples of the complexity of how different weather systems contribute to extreme wet seasons and section 6 provides a global overview of these contributions. Finally, Section 7 provides the summary and conclusions.

## 2 Dataset and methods

### 2.1 Identification of extreme wet seasons

We use daily accumulated precipitation fields from the ERA-Interim (ERAI) reanalysis of the European Centre for Medium-Range Weather Forecasts (Dee et al., 2011) for the period of 1979-2018, on a global grid with $1°$ spacing in both longitude and latitude. This is a rather short climatological period to analyse extreme wet seasons, providing roughly 40 precipitation seasons in most regions of the globe (Bombardi et al., 2017). Our overarching objective is to provide insights into their link

with weather systems. Therefore, "extremeness" is used in this study as a term with an impact-related content, rather than to characterize wet seasons statistically as periods with a low probability of occurrence. Extreme wet seasons (in the following just referred to as extreme seasons) have been defined separately at every grid point, as the consecutive 90-day period with the highest amount of accumulated precipitation in the 40-year period of 1979-2018. Prior to identifying extreme seasons, daily precipitation amounts less than 1 mm have been set to zero. This was done to avoid characterizing as wet days very low

model-produced accumulations. Choosing any consecutive 90-day period instead of the standard astronomical definition of seasons was motivated by variations of well-defined rainy seasons at different latitudes (Bombardi et al., 2019). However, considering only the top 90-day period of accumulated precipitation risks to neglect other periods that present almost equally-high precipitation amounts. Such periods are within the scope of this study, i.e. to relate weather systems with anomalously high seasonal accumulations of precipitation. Therefore, secondary extreme seasons have been also considered

at every grid point even if these seasons may not be statistically considered as extreme.These secondary seasons correspond to 90-day periods with accumulated precipitation exceeding 90% of the precipitation in the primary extreme season at the same grid point. All primary and secondary extreme seasons at one grid point were forced to not overlap in time. The result of this first step is, for every grid point, a list with the primary extreme season and between zero and 28 secondary extreme





seasons. Each of these seasons is characterized by their time period and precipitation amount. All results in section 3 are
based on this dataset.

## 2.2 Spatial and temporal coherence of extreme precipitation seasons

After identifying primary and secondary extreme seasons at every grid point, we defined their spatial coherence. To this end,
we consider that two neighbouring grid points are experiencing the same extreme season if their corresponding 90-day
periods overlap temporally by at least 75%, i.e. if they have at least 68 days in common. Figure 1 illustrates an example of
our approach for an idealized one-dimensional space-time grid, where extreme seasons with differing time periods have been
identified at six neighbouring grid points. According to our methodology, the extreme seasons identified at grid points 1, 2
and 3 fulfil the time overlap criterion and form a spatially coherent extreme season, which we refer to as a "patch" in the
following. Analogously, grid points 4 and 5 form an extreme season patch, but this patch is distinct from the patch formed by
grid points 1-3. Note that as an effect of this approach, a patch eventually extends over a time period that is longer than 90
days; we will address this issue in detail in section 4.2. Because every grid point may have several secondary extreme
seasons, the same grid point can be part of several patches. To identify all possible patches, we repeated the procedure
illustrated in Fig. 1 using as starting point every identified extreme season at every grid point (the primary and any secondary
ones). This resulted in an extremely high number of patches, with many (almost) identical patches. After removing
duplicates, i.e. patches with at least 90% of common points in space and time, we ended up with a total of 3734 patches, each
representing a spatially and temporally coherent extreme season. For all patches, the coordinates of their grid points and their
time periods are available as supplementary material.

## 2.3 Relating weather systems with extreme seasons

The relationship between extreme wet season patches and individual weather systems is examined for cyclones, warm
conveyor belts (WCB), tropical moisture exports (TME), and Rossby wave breaking (RWB). All these weather systems are
objectively identified in the 40-year ERAI dataset using six-hourly atmospheric fields and the methods described in Sprenger
et al. (2017) and references therein. In essence, at every 6-hourly time step of ERAI and for every weather system, the
algorithms identify spatially coherent clusters of grid points that belong to the same weather system, very much like the
patches of the extreme wet seasons. Table 1 provides a summary of the identification criteria and algorithms used. We
consider as RWB either a filamentary streamer or detached cut-off system of stratospheric potential vorticity. All four
weather systems are known to be related to heavy precipitation, either directly, e.g. by reducing static stability and favouring
convection, as in the case of stratospheric potential vorticity streamers and cut-offs or by strong lifting of moist air as in
WCBs; or indirectly by favouring dynamical processes that synergistically lead to spatially organised precipitation (cyclones
or TMEs). A common framework has been applied to quantify the contribution of these weather systems to extreme season
patches by considering the spatial overlap of the weather systems and of the patches, as explained further in section 4.2.





## 3 Statistical characterization of extreme precipitation seasons

Figure 2a shows the number of extreme seasons identified at each grid point, while Fig. 2b shows, as a reference, the global distribution of annual mean precipitation during the 40-year period in ERAI. Most regions, in particular most land and climatologically drier regions, show no more than one to five extreme precipitation seasons. However, the number of identified extreme seasons increases to 5-20 in areas where the annual precipitation amount is high, in particular in the inter-tropical convergence zone (ITCZ) and the mid-latitude storm tracks over the eastern North Pacific, the North Atlantic and in the Southern Ocean along 60ºS. This suggests that in these regions, seasonal precipitation typically varies only by fractions rather than multiples of the climatological mean. Therefore, numerous 90 day periods fall within our definition of secondary "extremely wet seasons". It is thus clear that in these regions our method identifies some periods that cannot be considered "extreme" from a statistical point of view, i.e., a period with a very low probability of occurrence. Yet, these periods reach almost the same accumulated precipitation as the locally wettest period and, therefore, we choose to use the terminology "extreme wet seasons" also for these periods throughout this manuscript.

Figure 3 shows the seasonality of the primary extreme seasons. Colour assignment is done according to the half of the month that includes the central date of each primary extreme season. In both hemispheres there is a clear shift in seasonality from oceanic to land regions. Over continental areas, extreme wet seasons occur mainly during boreal and austral summer, when convection triggered by strong solar radiation becomes important (see, e.g., Rüdisühli et al., 2020, for Europe). Over mid-latitude maritime areas, the extreme seasons occur mainly in boreal and austral winter, when storm tracks are fully developed and extratropical cyclones tend to me most intense. In regions where tropical cyclones occur frequently (e.g., in the Caribbean and southern Indian Ocean; the wettest seasons occur in the respective autumn season. For the Arctic, extreme seasons occur in late summer and early autumn when sea ice coverage is at its minimum, for Antarctica, however, the pattern is very heterogeneous. In the tropics, extreme seasons are most frequent during summer, following the latitudinal displacements of the ITCZ. However, Fig. 3 shows that the land-sea distinction is not equally sharp in all regions. For instance, the west coast of the US, the Iberian Peninsula and the north African coast, as well as Chile and eastern Australia all experience primary extreme wet seasons in winter. This suggests that such regions are strongly influenced by winter systems such as atmospheric rivers and cyclones (Rutllant and Fuenzalida, 1991; Leung and Qian, 2009; Lavender and Abbs, 2012; Flaounas et al., 2017). Other exceptions from the dominant summer occurrence of extreme wet seasons over land are several regions in the northern hemisphere where extreme seasons occur in spring, in contrast to summer for their neighbouring continental areas. This is especially observed near Iran, in the southern part of the Arabian Peninsula, and in eastern China and the eastern US. Especially for the east coast of US, spring time extreme seasons are plausibly related to anomalously high occurrences of daily extreme precipitation events (Li et al., 2018).





Next, the extreme seasons are statistically characterized. To this aim, Fig. 4a shows the ratio of precipitation amounts during these seasons to climatological, i.e., 40-year averaged values for the same 90 days. Only primary extreme seasons are presented, while results are similar for secondary seasons. For instance, if a grid point experiences two extreme seasons, one

from 10 Feb to 09 May 1991 and the second one from 23 Feb to 22 May 2012, then the value in Fig. 4a corresponds to the ratio of the average of the precipitation in these two specific periods with the average of the precipitation in all periods in the 40 years from 10 Feb to 09 May and 23 Feb to 22 May. By definition, extreme seasons have higher precipitation amounts than the climatology and therefore the amount ratio is everywhere larger than 1. However, Fig. 4a shows that this ratio strongly varies from close to 1 to more than 6. Comparison with the climatology in Fig. 2b shows that lower ratios are found

in areas where annual precipitation is high, such as within the ITCZ (where annual precipitation exceeds, on average, 2500 mm; see Fig. 2b) and along the mid-latitude storm tracks (roughly between 30° and 60° latitude in both hemispheres, where averaged annual precipitation in Fig. 2b is of the order of 1500 mm). These low precipitation amount ratios are consistent with the high numbers of extreme seasons in these regions (Fig. 2a). In contrast, high ratios of precipitation amounts are observed in areas where annual amounts are low, such as near the poles and in the arid subtropical areas along 30° latitude in

both hemispheres. The latter areas are climatologically affected by the descending branch of the Hadley cell, typically inhibiting precipitation occurrence and, therefore, an anomalously high seasonal precipitation amount has the potential of exceeding climatological values by a large factor. Finally, Fig. 4a shows that areas characterized by extreme seasons with amount ratios between 2 and 4 are located between strongly contrasting regions in terms of annual precipitation amounts (Fig. 2b). It is in these regions where spatial anomalies in the occurrence of precipitating weather systems (e.g., due to

anomalous cyclone tracks) may play a crucial role in forming extreme seasons. This will be discussed in more detail in section 6.

To gain more statistical insight into the factors that lead to extreme seasons, Figs. 4b and 4c show the ratio of extreme daily precipitation events and wet days in extreme seasons with respect to climatology (evaluated for the same 90-day periods as

the extreme seasons but in all 40 years). Daily extremes are defined individually at each grid point as daily precipitation values exceeding the 98th percentile of all wet days in the 40-year dataset, while wet days are defined by daily accumulations that exceed 1 mm. Comparing Figs. 4b and 4c, it is obvious that the ratios of daily precipitation extremes and wet days seem to show a contrasting pattern: a high ratio of daily precipitation extremes tends to co-occur with a low ratio of wet days, and vice versa. This is especially evident in areas of contrasting precipitation amount ratios in Fig. 4a. For

instance, in the ITCZ where precipitation is climatologically very frequent, an extreme season may only occur due to increased rainfall amounts. This is reflected in the anomalously high ratios of daily precipitation extremes in extreme seasons (Fig. 4b). In contrast, in arid areas where rainfall occurs rarely, few more wet days than climatologically can be responsible for a dramatic increase of seasonally accumulated precipitation. It is thus plausible that the lower the climatological precipitation amounts in an area, the more an extreme season is characterized by an anomalously high frequency of wet days.

On the other hand, in climatologically wet regions, extreme seasons are related to an anomalously high frequency of daily





extremes. Apart from this contrast between climatologically wet and dry areas on the globe, some regions have relatively high ratios of both daily extremes and wet days. Indeed, when comparing Figs. 4b and 4c, areas with a high ratio of daily extremes are spatially less constrained than areas with a high wet-day ratio. This is especially true in the tropics and mid-latitudes (up to 60º of latitude), suggesting that daily precipitation extremes may play a more widespread role for the occurrence of extreme wet seasons than the number of wet days.

In both Figs. 4b and 4c, the grand majority of ratios exceed the value of 1, suggesting that an extreme season typically occurs if there is a combination of both more wet days and more extreme events compared to the seasonal climatology. Indeed, Fig. 5a shows the probability density functions of the ratios of daily precipitation extremes and of wet days for all extreme seasons at all grid points. Clearly the spread of ratios of daily extremes is larger than the spread of ratios of wet days, with values between 1 and 5 and a median of 2.3 for daily extremes and a much narrower distribution with a median of 1.3 for wet days. Interestingly, the distribution for the daily extremes is bimodal with peaks near values of 1 and 2, respectively. The first peak is related to arid areas, where the 98th percentile of precipitation is close to zero. To combine information provided by the two ratios (mean values shown in Figs. 4b and 4c) and their variability (shown in Fig. 5a), we subjectively defined three ranges for the two distributions in Fig. 5a. These ranges are delimited by the peaks and the 75th percentile of the distributions (depicted by dashed lines in Fig. 5a). This forms a total of nine bins that serve to characterize each grid point according to the ratios of daily extremes and wet days required to form an extreme season (Fig. 5b). For instance, equatorial Africa and the Sahara are two contrasting regions of frequent and scarce precipitation, respectively. Cyan colour in equatorial Africa indicates a low wet day ratio of less than 1.2 and an intermediate daily extreme ratio between 2 and 3. Therefore, in this region, an extreme season requires only slightly more wet days than in the climatology but at least 2 to 3 times more daily extremes.

Despite the high spatial variability in Fig. 5b, several regional patterns can be distinguished. Areas related to high precipitation amounts (Fig. 2b) and a large number of extreme seasons (Fig. 2a), such as the storm tracks, are depicted by red colours in Fig. 5b (e.g. along 60ºS). These areas are characterized by wet day ratios of less than 1.2 and daily extreme ratios of less than 2. As discussed before, the identification of a high number of extreme seasons makes it difficult for these seasons to strongly exceed climatology. Other regions that experience high precipitation amounts due to the ITCZ have a daily extreme ratio exceeding 3 and a low wet day ratio of less than 1.2 (cyan colour), in agreement with the previous discussion of extreme seasons in this region. Extreme seasons with high wet day and daily extreme ratios (purple colours) mostly occur in the transition between areas of high and low climatological amounts of precipitation (Fig. 2b), for instance in subtropical maritime areas in both hemispheres(e.g. the eastern Atlantic and Pacific Oceans), but also in the eastern tropical Pacific. Especially the latter experiences major El Niño–Southern Oscillation (ENSO) events, which lead to a strong increase of wet days and daily extremes. Continental regions in the mid-latitudes are mostly characterized by green and orange colours, suggesting that extreme seasons are characterized by 20 to 60% more wet days and less than three times more daily extremes





than in the climatology. It is however noteworthy that several coastal areas have extreme seasons characterized by the highest ratios of wet days and daily extremes (purple colours), as for instance Portugal, Australia and Greenland.

## 4. Portrayal of extreme wet season patches

### 4.1 The 100 largest patches

The methodology to build patches of grid points with coherent extreme seasons (see Section 2.2) has been applied to all
primary and secondary extreme seasons. Figure 6 shows the 100 largest patches in three panels to avoid overlapping. In these panels, patches are labelled by the month and year of the average date of all 90-day periods that compose the patches. The size of the patches in Fig. 6 varies between $17.5 \times 10^5$ km$^2$ and $186 \times 10^5$ km$^2$. The largest one occurred from January to March 1983 in the eastern tropical Pacific (Fig. 6a).

Several patches correspond to or include well-documented periods of anomalously high precipitation, and some of them also reflect singular weather events that produced enough precipitation to characterise a whole 90-day period as an extreme season. For instance, the patches in the western North Atlantic in Figs. 6a and 6c depict the anomalously active hurricane seasons of 2010 and 2017. In contrast to these active hurricane seasons, the patch in the same region in September 1988 (Fig. 6b) does not correspond to one of the most active hurricane seasons, but rather includes the track of the extremely intense
Hurricane Gilbert (1988), the second most intense hurricane ever documented with a central pressure of 888 hPa. Other patches in the tropics agree with major El Nino and La Nina phases, such as the ones in 1982/1983, 1997/1998 and 2015/2016 in the central Pacific (Figs. 6a, 6b and 6c), and with ENSO-related extreme wet seasons in austral summer 2010/2011 (Fig. 6c; Ratna et al., 2014).

Within the storm tracks of the Northern Hemisphere, Fig. 6b shows two patches that are associated with anomalous variability of the polar jet: the first one corresponds to the extremely wet winter in the UK in 2013/2014, when an anomalously strong and persistent jet stream led to a series of extratropical cyclones hitting the region (Davies, 2015; McCarthy and Spillane, 2016). The second patch in the central North Atlantic along 35°N in winter 2009/2010 is associated with an anomalous southward deviation of the North Atlantic jet that led to a high frequency of TMEs and enhanced
precipitation over the western Mediterranean (Harnik et al., 2014; Sprenger et al., 2017). Other patches that depict known cases affected northwestern Australia in 2000 and 2011 (Figs. 6a and 6c), associated with enhanced cyclone activity and a strong Mascarene high (Feng et al., 2013). Another example is discernible in the eastern Antarctic where anomalously high precipitation occurred in autumn 1980 (Fig. 6a) as discussed by Van Ommen and Morgan (2010). All these examples provide insight into the variability of the specific weather systems and/or climatological features that can lead to extreme wet
seasons. This relationship will be analysed more systematically in the following sections.





### 4.2 Four example patches and definition of their core period

Before we can attribute the occurrence of weather systems to individual extreme season patches, we have to reconsider the temporal dimension of the identified patches. Because of our approach to build patches (see section 2.2), patches with many grid points might extend over a significantly longer period than 90 days. Such patches tend to be located in regions where
many secondary extreme seasons were identified, such as in the Southern Ocean. This is plausibly due to a higher likelihood of extreme season periods at neighbouring grid points to fulfil the temporal overlap criterion (section 2.2) in regions where there are more extreme seasons. In order to make the attribution to weather systems comparable across patches, the aim is here to define for each patch a "core period" that contains most of the area-integrated precipitation.

To illustrate this approach, we show detailed information about four selected example patches (labelled as a-d in Fig. 6) in Fig. 7. The central date, latitude and longitude of these patches are shown at the bottom of each panel. Figure 7a provides information about an elongated, tongue-like-shaped patch that affected the US west coast in winter 1992/93 (label a in Fig. 6a). Figure 7b corresponds to a rather large patch that covers parts of Australia in summer 2010/11 (label b in Fig. 6c). Figure 7c is for a patch in the Arctic in late summer 2016 (label c in Fig. 6a), and Fig. 7d presents an example in the Asian
summer monsoon region in 1991 (label d in Fig. 6b). The time period in each panel of Fig. 7 spans the earliest day (referred to as day 1) and the latest day (e.g. day 200 in panel a) from all 90-day extreme season periods that contribute to the considered patch. For each patch, three time series are shown in the panels of Fig. 7: (i) the upper graphs show time series of daily precipitation sum over all grid points that include the same day within their corresponding 90-day extreme seasons; For instance, let a certain day in the abscissa to be included in the 90-day extreme seasons of 15 out of 30 grid points that
compose a patch. Then the upper graph of Fig. 7 shows the sum of daily precipitation in these 15 grid points for that certain date. (ii) The middle graphs shows what we call the "percentage of contributing grid points", i.e. the percentage of grid points of the patch that contain the considered day in their 90-day extreme season period; and (iii) the bottom graphs indicate the occurrence of weather systems, as discussed below. For instance, the patch in the western US has a peak of area-integrated precipitation of ~23x10$^{12}$ litres on day 52. This value corresponds to the sum of daily precipitation from all grid
points of the patch (the percentage of contributing grid points is 100%), i.e. this day is included in all 90-day extreme season periods of the grid points that compose this patch.

This visualization is now helpful to explain how a "core period" can be determined for each patch. All four examples show that at the beginning and near the end of a patch period, only few grid points contribute to the patch and the area-integrated
precipitation values are lower in these periods. Also, in all cases, there is a more or less central time interval when (almost) all grid points contribute to the patch, and in these intervals the integrated precipitation is largest. We therefore define the core period as the longest period during which at least 25% of the respective grid points contribute to the patch. Considering again the western US patch (Fig. 7a), the so-defined core period extends from day 8 to day 104; for the monsoon patch (Fig.





7d) it becomes much longer from day 6 to day 141. Therefore, core periods of patches may last longer than 90 days, i.e. the
default time period that was initially used to define extreme seasons at individual grid points. In fact, further analysis shows
that for all 3734 patches, the median and the 75th and 95th percentile values of the core period durations amount to 99, 114
and 147 days, respectively. Assigning a flexible core period duration to each patch allows extreme wet season patches to take
into account the climatological characteristics of the different regions on the globe. For instance, core periods in the tropics
(Fig. 7d) may last for more than 100 days, corresponding to the duration of an intense monsoon season.


Finally, we now investigate the occurrence of the four objectively identified weather systems (Table 1) during the core
periods of the four example patches. The bottom graphs in the panels of Fig. 7 show coloured lines for each weather system
type indicating the days when a system overlaps with parts of the patch. Three shadings of colours are used to indicate
whether 5 to 33% of the grid points of the patch overlap with the weather system (light shading), or whether this percentage
amounts to 33–66% (medium shading), or to more than 66% (dark shading). For instance, dark green bars in Fig. 7a denote
days when TMEs overlap with more than 66% of the western US patch (especially WCBs never exceed 33% in Fig. 7 and
thus only light shading is visible). This provides qualitative information about the occurrence and relevance of a weather
system to the precipitation in the patch. Indeed, all four identified weather systems are known to be climatologically highly
relevant for heavy rainfall. It is however noteworthy that large patches may exhibit several local maxima of precipitation on
a given day, but not all of them necessarily overlap with one of the weather systems. In the following, we further investigate
the four exemplary patches of Fig. 7 to better understand the contribution of the four weather systems to the precipitation in
these extreme season patches.

## 5. Examples of how weather systems contribute to extreme wet seasons

### 5.1 Cold season patches in the subtropical-mid-latitude transition zone

The timeseries for the subtropical-to-mid-latitude patch in Fig. 7a exhibits distinct peaks in the 96-day core period. Several
of these peaks coincide with cyclones and TMEs, as shown in the bottom graph by the red and green lines, respectively.
Figure 8 provides insight into the complex relationship between precipitation, cyclones and TMEs for this patch, but also for
another patch of similar latitudinal extent and orientation that affected the Iberian Peninsula in late autumn 1989 (not shown
in Figs 6 and 7). For both cases, Figs. 8a and 8b show accumulated precipitation during the extreme season core periods of
200 to 550 mm with several local maxima. Local maxima in the northern parts of the patches correspond to regions where
cyclones occurred frequently (10 to 20% of the core period). On the other hand, the southern parts of both patches are co-
located with the northern extension of frequent TME occurrences (~20% of the core period). It is thus seems that cyclones
are the main contributors of precipitation to the patches, enhanced by the high availability of water vapour due to TMEs.
Blue lines in the lower graph of Fig. 7a coincide with prominent peaks of area-integrated precipitation in the US patch, for
instance on day 40. Hence, despite their non-frequent occurrence, WCBs may significantly contribute to seasonal





precipitation amounts. To underline this point, Figs. 8c and 8d show daily precipitation, sea level pressure, and the spatial extent of TMEs and of ascending WCBs for peak precipitation days during the US patch (day 40, i.e. 29 Dec 1992) and the Iberian patch (26 Dec 1989), respectively. In both cases, the local maxima of daily precipitation exceeding 50 mm coincide with WCBs in the warm sectors of deep cyclones. Such amounts represent large contributions to the total precipitation

during the patches' core period and thus the two examples highlight the important link between processes on the weather timescale (extratropical cyclones and their associated WCBs) and seasonal-scale extreme precipitation. In fact, Pfahl et al. (2014) showed that both these extreme wet season patches are in areas where cyclones and WCBs contribute frequently to intense precipitation (their figure 8). However, the two exemplary cases also show that cyclone and WCB-related precipitation can in turn overlap with both the patches in Fig. 8c and 8d. The synergy of these processes is responsible for

classifying these periods as extreme wet seasons. Finally, most cyclones co-occur with RWB events (Fig. 7a). However, this comes as no surprise since subtropical cyclones are typically instigated by the equatorward extensions of upper-level stratospheric filaments.

### 5.2 Warm season patch in the tropical-subtropical transition zone

Figure 9 provides more information about the northern Australian patch in summer 2010/11, previously introduced in Fig.

7b. This patch covers large parts of the maritime areas northwest of Australia and includes a tongue-like extension to the centre of the continent. Large parts of the patch overlap with areas with high cyclone frequencies, especially close to the west coast at 115ºE, 20ºS. However, the northern part of the patch is located within the ITCZ during the Australian summer. In this region, Coriolis forces are too weak for cyclones to develop. However, RWB events occur along the northern part of the patch during 60% of the patch's core period. Such upper-tropospheric systems can significantly contribute to the

formation of precipitation by reducing the static stability beneath and forcing vertical ascent. Further analysis showed that the narrow continental tongue of the patch is associated with a specific event: the landfall of tropical cyclone Yasi (track is shown in Fig. 9). Yasi make landfall at the Australian east coast and moved into the continent in February 2011, contributing strongly to the precipitation peak around day 165 in Fig. 7b. Consequently, this Australian patch has been formed through the combined contribution of climatological features (the ITCZ), enhanced precipitation by RWB, but also due to the frequent

occurrence of cyclones in the northwest of Australia plus the single, prominent system of tropical cyclone Yasi. Therefore, as illustrated in Fig. 9, patches should not be regarded as spatially coherent, in the sense that their spatial extension is due to a combination of specific weather systems such as tropical cyclones and climatological features such as the ITCZ.

### 5.3 The summer 2016 patch in the Arctic

Figure 7c depicts a large patch that covers the eastern Arctic in late summer 2016 (dotted area in Fig. 10). Figure 10 shows

that a large part of this patch is related to an anomalous occurrence of cyclones, with frequencies more than 25% higher than in the climatology. The region with anomalously frequent cyclones (red contour) agrees with the southward extension of the patch into Siberia near 150ºE and with an anomalously high precipitation excess compared to climatology of more than 1.7.





Figure 7c shows that several prominent peaks in the precipitation time series coincide with WCBs and RWB events, similarly to the subtropical patch in Fig. 7a. The year 2016 has been recorded as the warmest in the last decades in the Arctic

and was characterized by anomalously low sea-ice extent and overall positive sea surface temperature anomalies that enhanced evaporation and consequently precipitation (Simpkins 2017; Overland et al., 2018; Petty, 2018). Such conditions led to the extreme season in the eastern Arctic and probably reflect a rainier future regime in the Arctic region (Bintanja, 2017).

### 5.4 Patch in the Asian monsoon region

The tropical patch in Fig. 7d covers large part of the Bay of Bengal and affects Southeast Asia (Fig. 11). Indeed, Fig. 11 shows a local maximum of accumulated precipitation over Myanmar of about 2700 mm. The rather long core period of this patch reflects the intense monsoon season in the region from June to mid-October. High values of area-integrated precipitation occur during the whole core period (Fig. 7d). In contrast to the other examples, few cyclones and TMEs overlap with the patch. In contrast, RWB events are frequently present mainly on the eastern side of the monsoon anticyclone (Kunz

et al., 2015; their Fig. 9a). Figure 11a shows that there is indeed a high percentage of identified RWB during the core period, mainly at the eastern side of the patch, within the latitude range of 5º to 20ºN. An example of a prominent RWB event is presented in Fig. 11b, which shows a tongue of high PV values (>1.5 PVU) extending from China into the Bay of Bengal at 10ºN. This streamer is co-located with several local maxima of daily precipitation within the patch, plausibly contributing to the formation of daily extreme events, a necessary aspect of extreme wet seasons in this region (Fig. 5b).

## 6. The contribution of weather systems to extreme wet seasons: a global view

Following the four examples of the previous section, we quantified the occurrence of the four objectively identified weather systems during all the 3734 identified patches. To this end, we counted the number of days with an overlap of each weather system with the patch during its core period. In order to then estimate whether these numbers are anomalous compared to climatology, this process was repeated for the same dates as for the core period, but for all 40 years of our dataset. The ratio

then defines the overlapping frequency ratio of a patch with respect to climatology and results are presented in Fig. 12. For instance, orange patches in Fig. 12a are overlapping about 2.5 times more often with cyclones during their core periods than in the climatology. In addition to all patches, the right-hand panels of Fig. 12 show the latitudinal distribution of the overlapping frequency ratios. They were calculated by zonally averaging the ratios of all patches within +/- 7.5º of each latitude degree.


Figure 12a reveals the importance of cyclones for the formation of precipitation in extreme wet seasons. The largest part of the world is covered by patches with frequency ratios of at least 1.2. The latitudinal distribution of these ratios shows two local maxima in the subtropics close to 30º latitude. Indeed, Fig. 12a shows that several patches in the subtropics have





cyclone frequency ratios of more than 2, and in some cases even more than 4. Patches with high ratios occur in particular in

subtropical oceanic regions, in transitional areas between climatologically high and low precipitation amounts (Fig. 2a).
These regions are also characterized by an anomalously high number of wet days and daily extremes in Fig. 5b. In
accordance with Pfahl and Wernli (2012), which implies that individual cyclone tracks that occur equatorward of the
climatological storm tracks may produce daily precipitation extremes and thus significantly contribute to extreme seasons.
Low ratios in Fig. 12a occur along the equator due to the absence of cyclones, but also within the storm track regions (e.g.

along 60ºS). Especially in the southern hemisphere, the right panel of Fig. 12a shows that ratios decrease monotonically from
1.45 at 30ºS, to 1.1 at 60ºS. This result suggests that the closer a patch is located to a climatological storm track, the more
unlikely it is for this patch to overlap with more cyclones than in the climatology. The same also holds in the northern
hemisphere in the western North Atlantic and in the eastern North Pacific between 30 and 60ºN.

WCBs are airstreams associated with extratropical cyclones that ascend along the cyclones' frontal surfaces. They are
associated with both stratiform and convective precipitation, and they can contribute significantly to extreme precipitation
events (e.g. Pfahl et al., 2014; Oertel et al., 2019). Since WCBs are directly related to the occurrence of cyclones, Figs. 12a
and 12b are expected to be similar. This is partly confirmed by the latitudinal distribution of WCB ratios with peaks in the
subtropics at 30ºS and 25ºN, similarly to the zonal averages of cyclone ratios. However, Figs. 12a and 12b also present

considerable differences. For instance, the frequency ratio of WCBs along 60ºS (Fig. 12b) is significantly higher than the one
of cyclones (Fig. 12a). This suggests that extreme seasons within the storm tracks are not formed due to a higher frequency
of cyclones but due to their physical characteristics. Indeed, a more frequent occurrence of WCBs contributes essentially to
the enhancement of seasonal precipitation and thus to the formation of extreme seasons in mid-latitude oceanic regions, but
also in continental and polar regions. However, it is noteworthy that the scarcity of WCBs especially in the polar regions

(e.g. Fig. 7c, Madonna et al., 2014) contributes to the high ratios in Fig. 12b, even if relatively few WCBs occurred during
the extreme season.

TMEs correspond to moist plumes that originate from the tropics and extend into the extratropics. Therefore, TMEs are
expected to favour high amounts of precipitation whenever they reach higher latitudes. Indeed, Fig. 12c shows several

patches with high TME frequency ratios, especially along 60ºS, but also in the continental areas of Asia and North America.
A quasi-constant zonal average of 1.1 is observed in the mid-latitudes (right panel of Fig. 12c), suggesting that TMEs may
contribute to the formation of extreme seasons in the extratropics. However, this contribution is expected to be weaker than
the one from cyclones and WCBs. Occasionally, TMEs contribute to Arctic extreme seasons, although as for WCBs, the high
ratio values there results from very few events during the extreme seasons and even less in the climatology.


Finally, RWB events can directly favour regional precipitation through reducing the static stability of the atmosphere, or
indirectly by inducing large-scale ascent and/or the formation of surface cyclones in mid- and polar latitudes (Wernli and




Sprenger, 2007). Figure 12d shows generally low values of overlapping ratios of RWB, rarely exceeding values of 1.5. This

is a consequence of the fact that RWB is climatologically frequent and thus the contribution of RWB to extreme wet seasons cannot be as significant as the one of cyclones and WCBs, two weather systems with a lower climatological frequency. The latitudinal profile of RWB frequency ratios (right panel of Fig. 12d) presents two local minima, both in the mid-latitudes of the two hemispheres where RWB is particularly frequent. However, when elongated RWB-related stratospheric filaments occasionally intrude into the tropics, then extreme precipitation may be triggered (Knippertz, 2007). Because such events are rare and intense, a maximum of RWB ratios occurs in the tropics. It is plausible that RWB into the tropics leads to daily

extreme events, a necessary ingredient for the formation of extreme seasons in these latitudes, as suggested by Figs. 4b and 5b and discussed in Section 5.4. It is noteworthy that other than the four systems that we objectively identified in this study might be also involved in forming daily precipitation extremes in the tropics (e.g. a strongest ITCZ or warmer seas surface temperatures). Finally, the relatively high frequency ratios in polar latitudes may be related to the high frequency ratios of WCB in polar latitudes (Fig. 12b), especially in the Southern Hemisphere. Indeed, WCBs are known to transport air masses

of low PV towards the higher troposphere, deepen the ridges in a poleward direction and significantly contribute to RWB (e.g. Grams et al., 2011; Madonna et al. 2014b).

## 7. Summary and conclusions

This study investigated extreme wet seasons globally, and introduced them as a new concept at the interface of weather and climate research. First, we defined primary extreme wet seasons separately at every grid point of ERAI as the 90-day period

with the largest accumulated precipitation in the last 40 years. To also account for 90-day periods with only slightly less precipitation, we also considered periods that exhibit at least 90% of accumulated precipitation of the primary extreme season and called them secondary extreme seasons. Our results show that the definition of extreme wet seasons becomes a delicate issue in some areas. For instance, at some grid points in the Southern Hemisphere storm tracks up to 20 extreme seasons have been identified. In these regions where the variability is small compared to the mean, 90-day accumulated

precipitation amounts show a rather small variability, rendering the label "extreme" for some of the identified seasons less meaningful from a statistical point of view. However, in many regions the identified extreme seasons exceed the climatologically expected precipitation amounts by large factors and several of them have been reported in the literature as particularly impactful.

Further analyses focused on the statistical characterisation of extreme seasons by counting the number of wet days and daily precipitation extremes during the extreme seasons. The grand majority of extreme seasons include both more wet days and more daily extremes than in the climatology. Nevertheless, these two metrics allow different precipitation regimes to be delineated with two contrasting scenarios: regions of scarce precipitation (e.g. arid regions) where extreme seasons may only occur due to just a few more wet days, and regions with frequent precipitation (e.g. along the ITCZ) where daily extremes

are pivotal for the occurrence of extreme seasons. Our results show that only extreme wet seasons in subtropical regions, i.e.





located in the transition between rainy and arid areas, include significantly more wet days *and* daily extremes. A methodology was next applied to concatenate extreme seasons at individual grid points to patches, where each patch is affected by the same extreme season. Large patches were related to planetary-scale events such as extreme El Niño years, or the ITCZ, but also to single weather systems such as major hurricanes making landfall in North America.


The importance of synoptic-scale dynamics for the formation of extreme seasons was also investigated by identifying four weather systems, known to be important for the (extreme) daily precipitation events, and by considering their overlapping frequency with the extreme season patches compared to climatology. To analyse and illustrate the complex interactions between these four weather systems and extreme season patches, we first investigated in detail four example patches in the

mid-latitudes, the subtropics, the Arctic and the tropics. Furthermore, we analysed the global distribution of patches along with their relationship to the four objectively identified weather systems. Results highlighted the anomalously high occurrence of cyclones as a crucial element for the formation of most extreme seasons, except in the tropics and the storm tracks where cyclones are either unlikely to occur or very common. However, our results showed that extreme seasons in the storm track regions are related to an anomalously high occurrence of warm conveyor belts. Finally, Rossby wave breaking

events have been found to contribute to the formation of extreme seasons at low latitudes, while tropical moisture exports were found to contribute to extreme seasons in the mid-latitudes and the sub-tropics. Consequently, weather systems of different latitudinal origin may be important for the formation of extreme seasons. In fact, our overall results showed that extreme wet seasons can be either related to a higher occurrence of similar weather systems (e.g. the hyperactive hurricane season of 2017), to the contribution of single exceptional events (e.g. the landfall of hurricane Yasi for the Australian

extreme season in summer 2011), and/or due to the influence of weather systems that are climatologically uncommon in specific regions (e.g. RWB in the tropics).

The variability of atmospheric dynamics within a 90-day period is large and therefore it is a challenge to perform a detailed quantification of the contribution of weather systems to extreme wet seasons. Nevertheless, our results show that valuable

conclusions can be reached by combining the objective identification of weather systems, together with a grid-point-based statistical analysis of precipitation. In this context, a high number of patches, especially the ones shown in Fig. 6, may be considered as interesting cases of strongly anomalous seasonal precipitation that merit further investigation. Therefore, future research could focus on such case studies to better determine the degree of complexity of the dynamics involved, to develop new statistical characterisations of seasonal precipitation, or even bridge more efficiently climate and weather

perspectives on extreme wet seasons.

**Data availability.** ERA-Interim data can be downloaded from the ECMWF web page at: https://apps.ecmwf.int/datasets/data/interim-full-daily/levtype=sfc/ (European Centre for Medium-Range Weather Forecasts, 2020).



**Author contributions.** EF and HW conceived the study and methods. EF wrote major part of the paper and performed the analysis. MS provided technical support and all co-authors contributed to writing the paper and commented on its earlier versions.

**Competing interests.** The authors declare that they have no conflict of interest.

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

**Tables**

| Cyclones | Grid points within the outermost sea level pressure contour enclosing one local minimum (Wernli and Schwierz, 2006) |
|---|---|
| Warm conveyor belts | Grid points overlapping with the ascending part (between 800 and 400 hPa) of air parcels that rise for at least 600 hPa within 48 hours (Madonna et al., 2014). |
| Rossby wave breaking | Grid points were either PV cutoffs or streamers are located.<br><br>PV cutoffs: Grid points with stratospheric air (PV > 2 PVU), detached from the main stratospheric body on any isentropic level between 305 and 370 K (Wernli and Sprenger, 2007).<br><br>PV streamers: Grid points within narrow filaments of stratospheric air on any isentropic level between 305 and 370 K (Wernli and Sprenger, 2007). |
| Tropical moisture exports (TME) | Grid points overlapping with 7-day forward trajectories started from the tropics (20ºS–20ºN) that reach 35º latitude in either hemisphere with a horizontal moisture flux of more than 100 g kg$^{-1}$ m s$^{-1}$ (Knippertz and Wernli, 2010). |

**Table 1** Short description of the six objectively identified weather systems.






## Figures

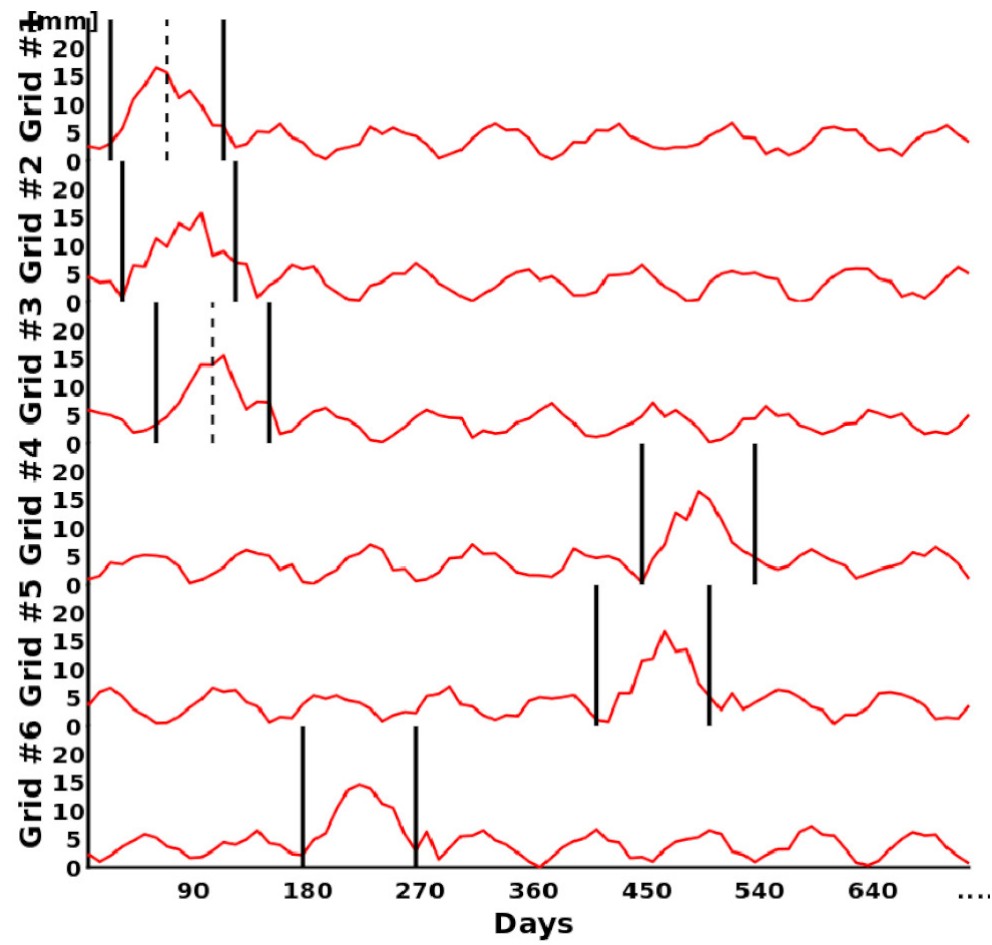

**Figure 1** Methodological approach in an idealized one-dimensional grid to identify spatial coherences of extreme seasons. Red lines show precipitation time series per grid point, vertical black lines delineate the identified extreme season per grid point and vertical dotted lines depict their central date (only for two seasons, to be used as an example in text).



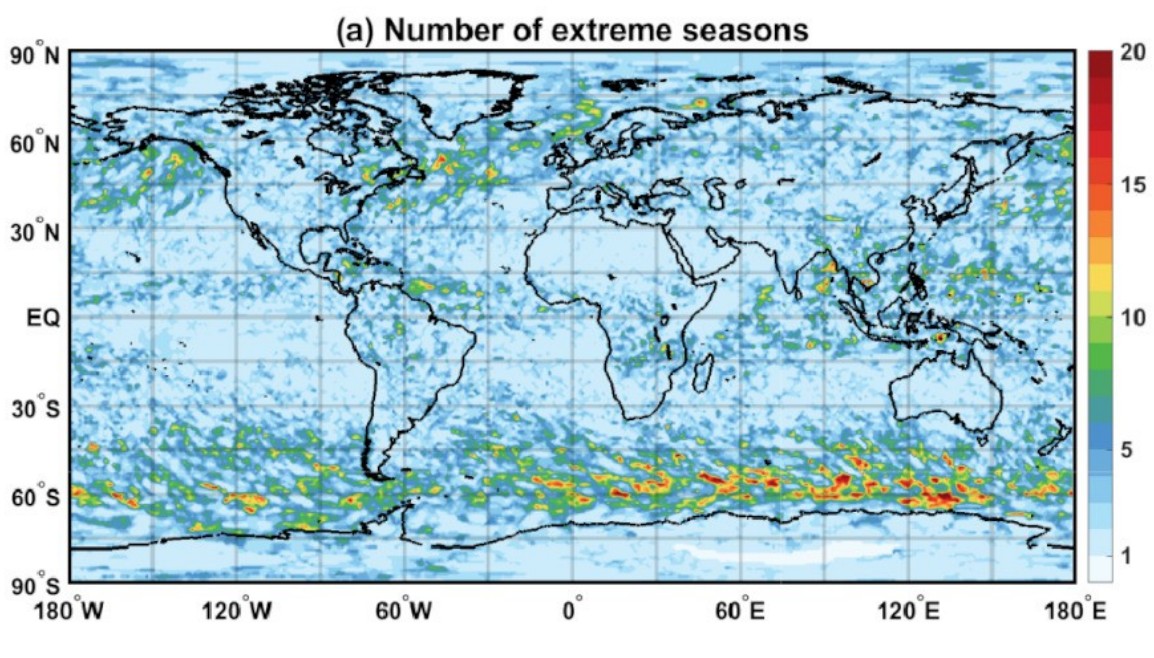

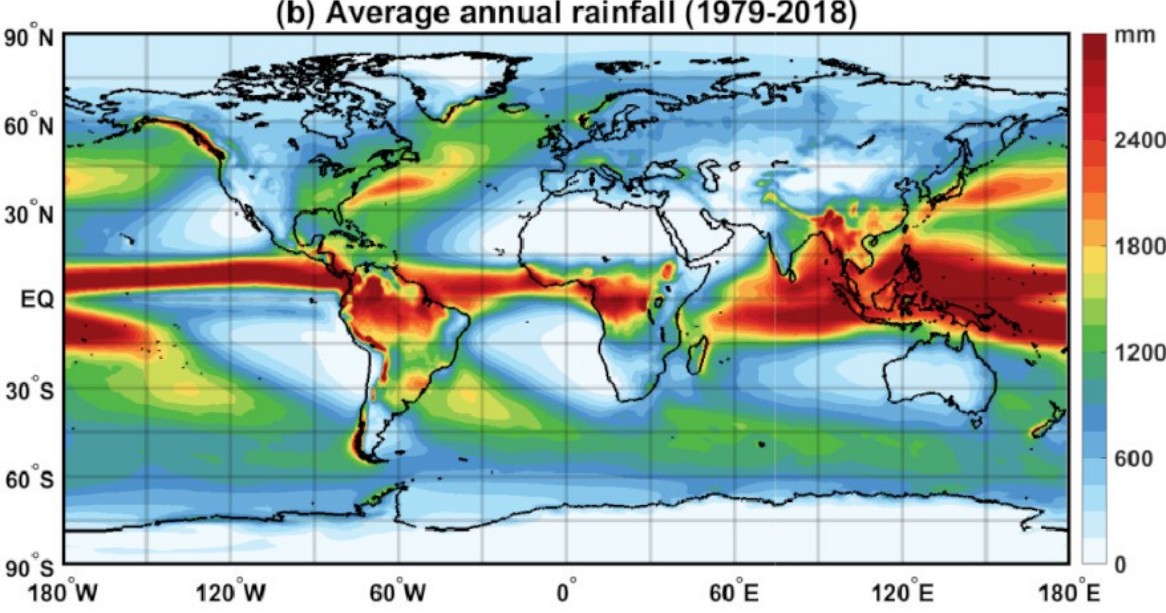

**Figure 2 (a)** Global distribution of the number of extreme seasons. **(b)** Average annual rainfall in a 40-year period (1979-2018).

770



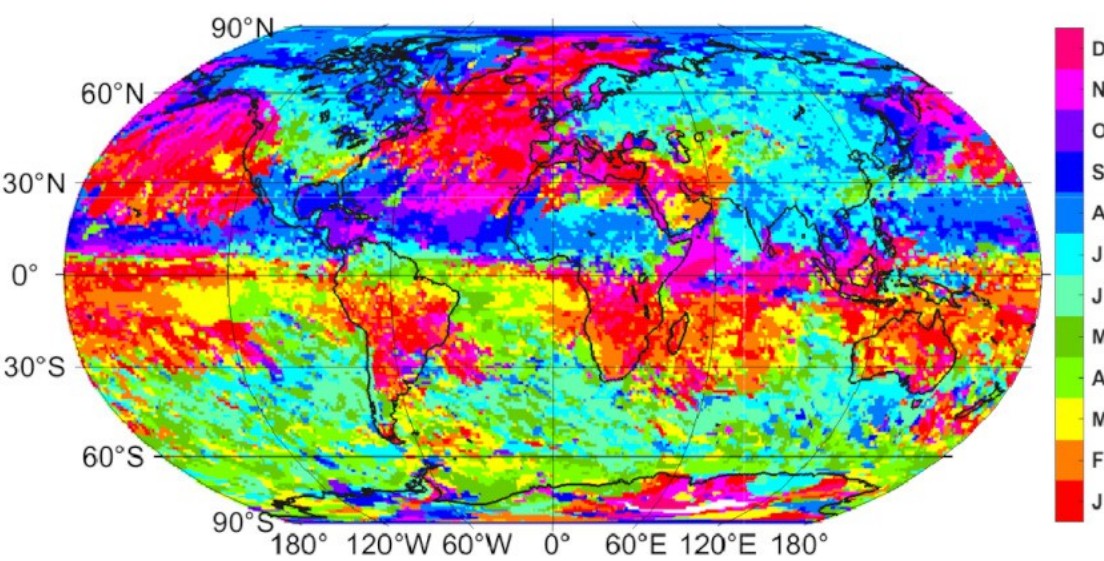

**Figure 3** Monthly distribution of the central day of primary extreme seasons.







**Figure 4 (a)** Ratio of precipitation amount of extreme seasons respect to seasonal average **(b)** number of daily precipitation
805    extremes included in an extreme season respect to the seasonal average. **(c)** as in **(b)** but for the number of wet days.





**Figure 5 (a)** Probability density function of number of ratios of daily precipitation extremes and precipitable days for all extreme seasons and for all grid points. Vertical dotted lines correspond to ratios of 1.2, 1.6, 2 and 3. **(b)** Attribution of grid points to combined climatological ratio fractions of number of precipitable days and daily precipitation extremes. Fractions are delimited by dotted lines in panel **a**.






**Figure 6** 100 largest patches, labelled with the central month and year of all included extreme season. For clarity reasons, all areas are distributed in three panels and are depicted by different random colours. Four patches are also labelled by a green letter that corresponds to panels in Fig. 7.



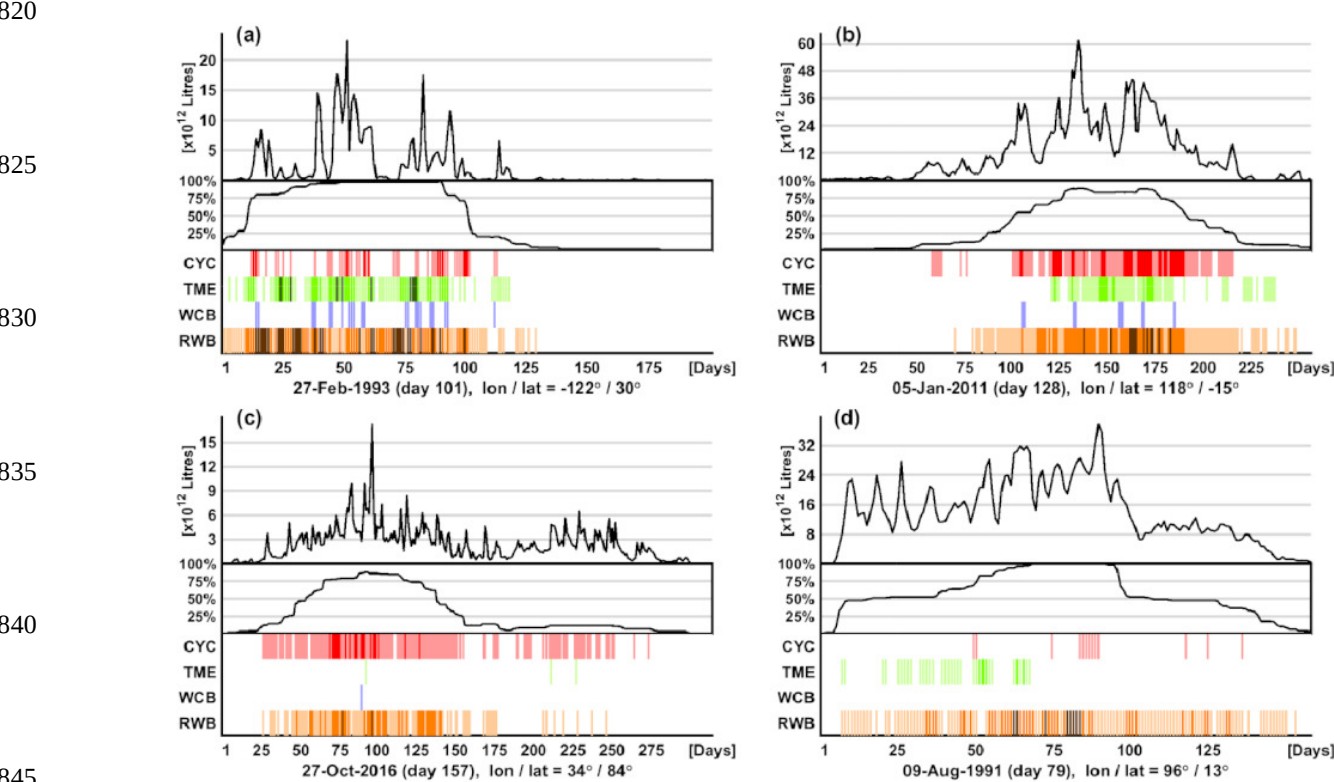

**Figure 7** Each of the four panels depicts an exemplary patch, labelled by a letter in Fig. 6. Time periods in abscissa span the earliest (day 1) and latest date (e.g. day 170 in panel A) of all extreme seasons of the grid points that compose each patch. The upper part of each panel shows time series of daily precipitation, accumulated for all grid points that compose the patch. Given that the patch period in abscissa is composed by non-identical extreme seasons per grid point, the time series in the middle of each panel shows the percentage of extreme seasons that include each day in abscissa. The lower part of the panel marks each day by a vertical line if a weather system overlapped with the patch (see text): red for cyclones, blue for WCBs, green for TMEs and brown for RWB. Central date and average latitude/longitude of each of the four patch is shown under the panels.



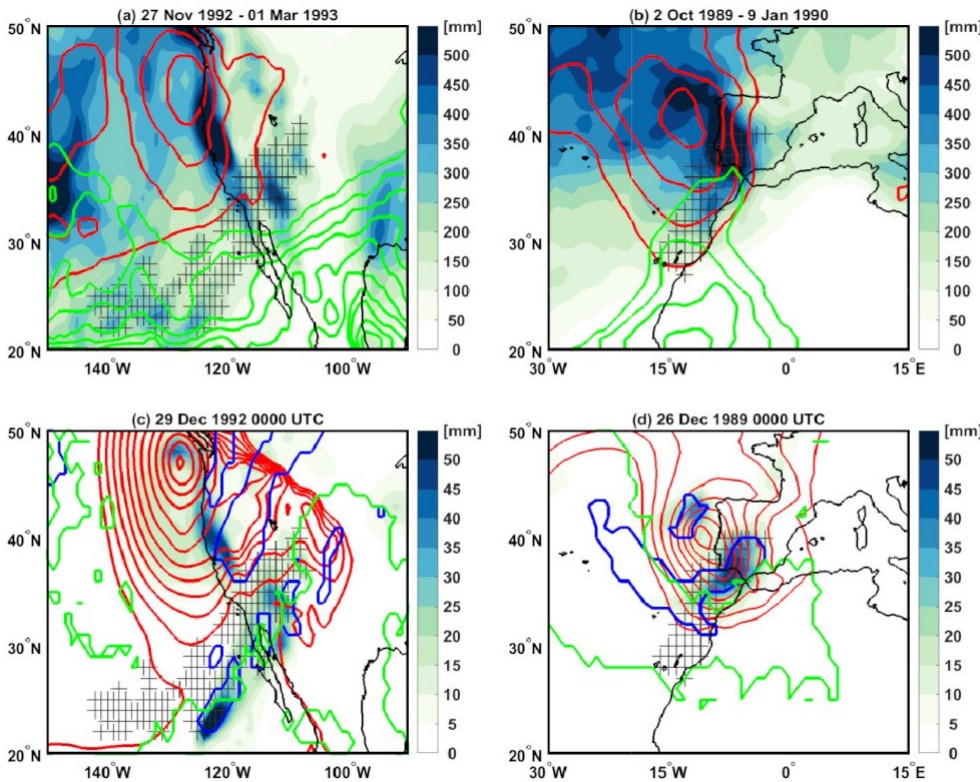


**Figure 8 (a)** Accumulated precipitation from 27 November 1992 to 1 March 1993 (in colour), cyclone frequency is shown in red contours (with an interval value of 5%, starting from 10%) and TME frequency is shown in green contour (with an interval of 5%, starting from 20%). Feature frequencies are calculated on the basis of six-hourly outputs from ERAI. **(b)** as in **(a)** but for different period and region. **(c)** 24-hour accumulation of precipitation from 28 December 1992, 1200 UTC to 29 December 1992 1200 UTC (in colour). Red contours show sea level pressure on 29 December 1992 0000 UTC (starting from 1015 hPa and with a step of -3 hPa). Green contour shows the areas co-locating with TMEs and blue contour shows WCB ascent objects. In all panels, the spatial extent of the patches are represented by the hatched area.






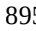

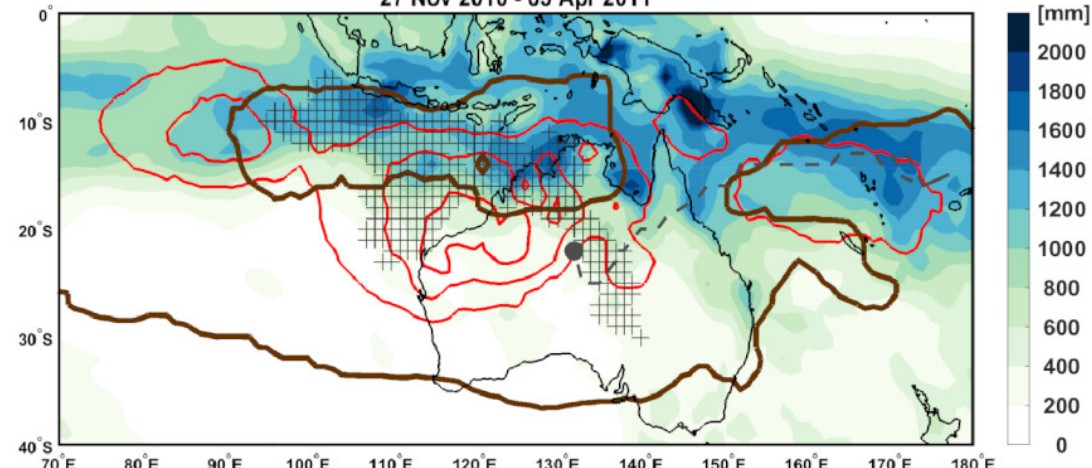




**Figure 9** Accumulated precipitation from 27 November 2010 to 3 April 2011 (in colour). The frequency of cyclones occurrence during this period is shown in red contours for values of 20, 40 and 60%, while the brown contour shows Rossby wave breaking frequency exceeding 60%. Feature frequencies are calculated on the basis of six-hourly outputs from ERAI.
The grey dashed line shows the track of tropical cyclone Yasi, while its position on 5 February 2011, 18 UTC, is represented by the red dot. The spatial extent of the patch is represented by the hatched area.

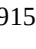

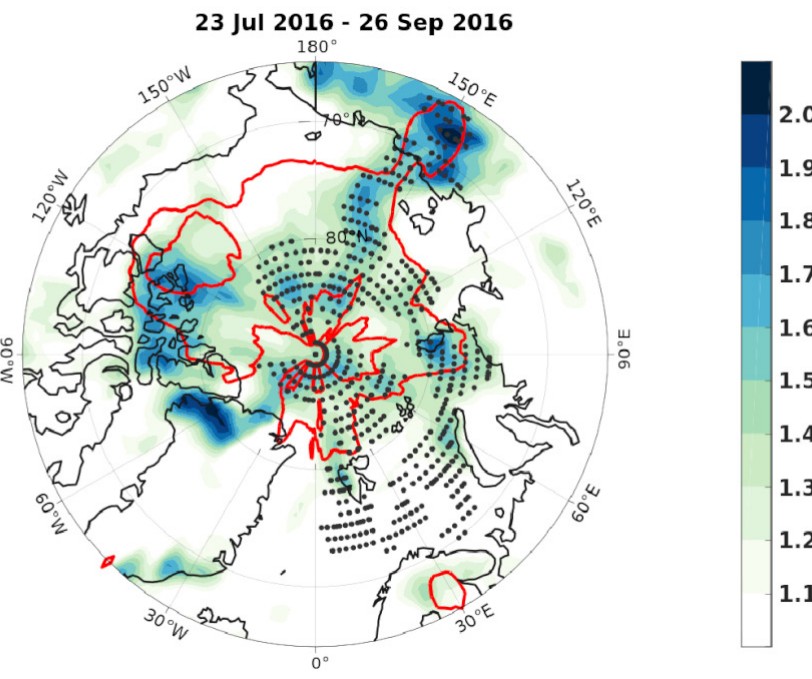

**Figure 10** Ratio of accumulated precipitation during the period 23 July 2016 to 26 September 2016 and the climatological
values of same dates (in colour). Red contours show the areas where the climatological ratio of cyclones occurrence is 1.25 and 1.5. The spatial extent of the patch is represented by the dotted area.

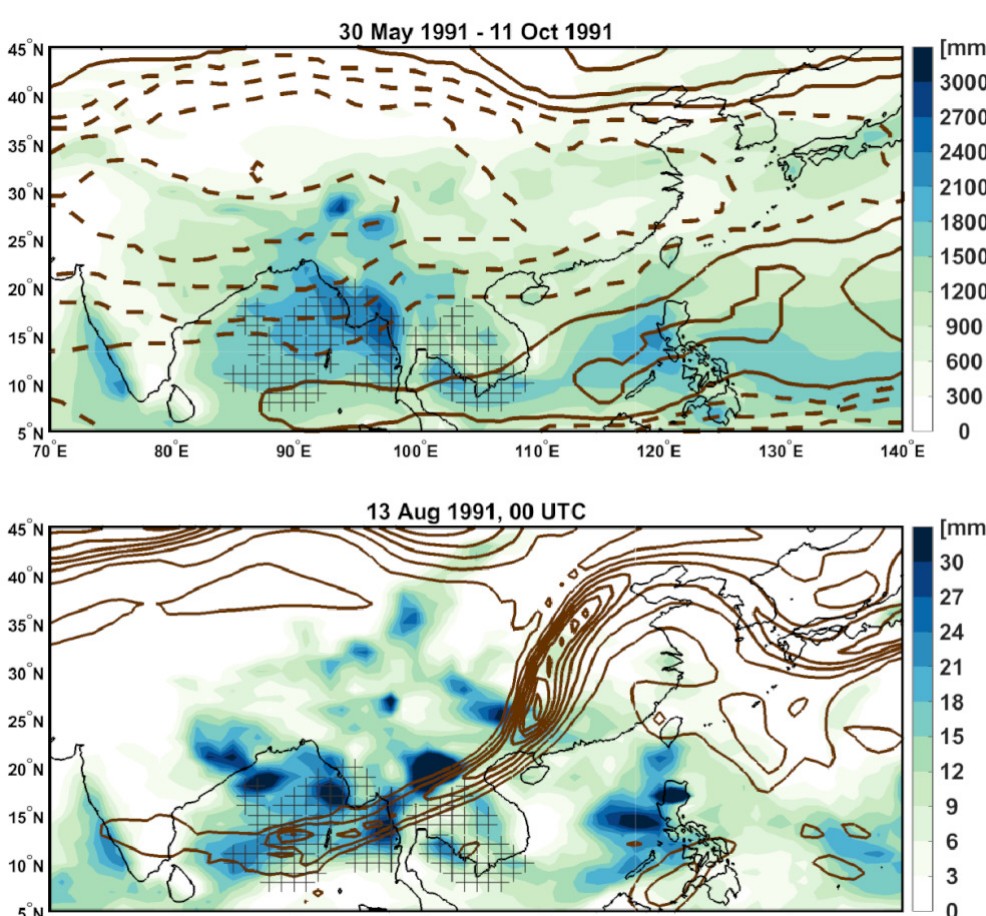

**Figure 11** Top panel shows the accumulated precipitation from 30 May 1991 to 11 October 1991 (in colour). Brown contours show the frequency of RWB occurrence during this period (contours interval is 10%; dashed contours show values of 40% or less, and solid contours are used for values of 50% or more). Feature frequencies are calculated on the basis of six-hourly outputs from ERAI. Bottom panel shows daily accumulated precipitation for 13 of August 1991 (in colour) and PV values of more than 1.5. PVU (with an interval of 1 PVU) at the potential temperature level of 360 K, at 00 UTC.



**Figure 12** All patches are coloured according to their overlapping frequency ratios with specific weather systems. Panels in the right column show the latitudinal distribution of the overlapping ratios, as zonal averages within +/- 7.5° in latitude. Patches may overlap between each other and thus illustration started from the patch presenting the lowest ratio.