# Peer review of "Extreme wet seasons – their definition and relationship with synoptic scale weather systems"

_Weather and Climate Dynamics, 2020_

## Referee Comment (RC2) · Anonymous Referee #2 · 13 Aug 2020

GENERAL COMMENTS

This study provides a novel analysis of what the authors refer to as extreme precipitation seasons, defined as 90-day periods during 1979–2018 exhibiting especially large precipitation accumulations. A global climatology of these seasons is constructed and their characteristics are examined through statistical analysis. Contemporaneous global climatologies of warm conveyor belts, tropical moisture exports, breaking Rossby waves, and cyclones are employed to examine dynamical processes that contribute to the extreme seasons.

Overall, I found this study to be interesting and novel, and I believe that the topic fits

within the scope of Weather and Climate Dynamics. The methods developed to identify extreme precipitation seasons and extreme season patches are innovative and novel, though, in my opinion, somewhat complicated. This is the first study to construct a global climatology of extreme precipitation seasons and to attempt to relate them to different types of weather systems. I believe that the study addresses important gaps in scientific understanding regarding the occurrence of extreme precipitation seasons. Despite the strengths of this study, there are a number of issues that need to be addressed with regard to the clarity of the writing, interpretation of the results, the methodology, and the background discussion.

SPECIFIC COMMENTS

Abstract: The abstract is quite lengthy and complicated. I recommend simplifying and shortening it.

line 43: I recommend being more specific regarding the socioeconomic impacts of these events.

line 47: I suggest also mentioning climatological studies of relationships between PV streamers/breaking waves and precipitation extremes (e.g., Martius et al. 2006; de Vries et al. 2018; Moore et al. 2019).

line 51: It is unclear what exactly you mean by 'environmental risks' in this context. Please clarify.

line 53: Specify what impacts the hurricanes caused and the coastal regions of the United States that they affected.

line 53–54: Note, however, that the season did include several extreme-rain-producing hurricanes.

line 54: Specify what the 'main impact' was? Was it prolonged regional flooding?

line 56: Please provide a reference for this statement.

[Figure]

line 111: Please explain why this model-based dataset was used. Also, please state any caveats that must be considered when using coarse-resolution model-based precipitation data.

line 155–158: In my view, the authors have not provided sufficient context and background information to motivate examination of relationships to these different weather systems. This sentence is inadequate in this regard and does not fully and accurately describe the influence that these systems can have on precipitation. For instance, the authors fail to mention that PV streamers and cut-offs have also been found to be linked to strong water vapor transports and dynamical lifting. The four weather system types and their dynamical relationships to precipitation extremes should be described in more detail in the introduction section. Also, it could be worthwhile to describe interrelationships between the four types of systems.

line 168: I suggest using a consistent term for the extreme seasons throughout the paper. Use either "extreme wet season" or "extreme precipitation season" but not both.

169–171: While I understand your justification for classifying these seasons as extreme, I am still unsure whether I agree with it. If the seasonal precipitation does not deviate much from climatology, then it really is indicative of an ordinary precipitation season. Are there ways to avoid inclusion of so many secondary seasons in the dataset? Could you use more restrictive criteria to identify secondary extreme seasons? Could you just consider the primary extreme seasons and not the secondary seasons?

line 178: Perhaps insert "and occur most frequently" after "most intense"?

line 184: "This suggests. . ." I do not see how a lack of a sharp land–sea distinction itself suggests that a given region is influenced by atmospheric rivers and cyclones. It would be more precise to say that the lack of a distinction suggests that a region is influenced by landfalling systems originating over the ocean, such as extratropical cyclones and atmospheric rivers.

line 194–197: Apologies for my confusion, but I am having trouble reconciling this sentence with the previous sentence. If only results for primary seasons are presented, then how can there be multiple extreme seasons at a given grid point.

line 222: "arid areas": I suggest providing specific examples of these areas to aid the reader.

line 225: "climatologically wet regions": I suggest providing specific examples of these regions to aid the reader.

line 273: Please provide references for the 2010 and 2017 hurricane seasons.

line 351: It is not clear to me how unusual the frequencies of cyclones, streamers, and TMEs depicted in Figs. 8 and 9 are for those regions and seasons. It would be helpful to compare the feature frequencies to the climatological frequencies for the time periods, as was done in Fig. 10.

line 363–364: "However, the two..." It is unclear to me what the purpose of this sentence is.

line 364–365: "The synergy..." The meaning of this statement is ambiguous to me. Which processes are you referring to?

line 365: "Finally, most..." Mention that this statement applies specifically to the 1992–1993 event.

line 365–367: "However, this comes..." What is the basis for this statement? Please provide a supporting reference.

line 373: "In this region..." This is not true. Cyclones can and do occur at these latitudes, as clearly depicted in Fig. 9.

line 373–374: "However, RWB..." A figure reference is needed in this sentence.

line 374: By "upper-tropospheric systems" do you mean elongated PV streamers associated with RWB? If so, consider saying "The upper-level PV streamers resulting from the events". Upper-level is more accurate than upper-tropospheric here given that these systems are defined as narrow filaments of stratospheric high-PV air.

line 389: The anomalous warmth could also reflect frequent poleward excursions of warm, moist air into the Arctic that supported the precipitation within the patch.

line 392: "probably reflect..." this assertion does not appear to be supported by any evidence.

line 416: What do you mean by "the largest part of the world"?

line 417: Does this imply that the cyclone climatology used in this study also includes tropical cyclones and other tropical low pressure systems in addition to extratropical cyclones? Is there any distinction made in the climatology between extratropical and tropical systems?

line 422–423: I find this sentence confusing. Which result is in accordance with Pfahl and Wernli (2012)? Also, it is a sentence fragment.

line 430: it would be more dynamically accurate to say "baroclinic zones associated with cyclones" instead of "cyclones' frontal surfaces"

line 437: "physical characteristics" is vague. Please specify the physical characteristics that are relevant in this context.

line 443–444: "Therefore, TMEs..." This statement strikes me as erroneous. Can you cite a study that supports this claim? My understanding is that a TME will only support heavy precipitation where it encounters a region of strong ascending motion; thus, TMEs should not be expected to produce high amounts of precipitation whenever they reach higher latitudes but rather only under certain circumstances.

line 448–449: "Occasionally, TMEs..." I find this sentence somewhat confusing. Please rephrase more clearly.

line 454–455: I do not entirely follow this reasoning. The ratios shown in Fig. 12 do not necessarily indicate the strength of the contribution of a given type of weather system. They only indicate the degree to which weather system frequencies deviate from climatology during extreme precipitation seasons. It seems to me that it is still possible for systems to produce large portions of the precipitation during extreme seasons even if their frequencies do not deviate substantially from climatology.

line 457: It would be more precise to say "PV streamers" rather than "filaments"

line 459: What do you mean by "RWB into the tropics"? Perhaps it would be more accurate to say "extension of PV streamers into the tropics".

line 461–463: "It is noteworthy that. . ." I really do not understand this sentence. Please clarify.

line 463–464: "Finally, the. . ." This sentence does not make sense to me.

line 464–466: "Indeed, WCBs. . ." I do not understand how this sentence connects with the preceding discussion in this paragraph.

line 488: It seems to me, based on the results in Figs. 8–11, that large patches can also result from synoptic-scale weather systems, such as extratropical cyclones and RWB. This should also be mentioned here.

line 499–500: The streamers that form in connection with wave breaking tend to be part of baroclinic waves that are tilted with height. Thus, widespread heavy precipitation produced in association with wave breaking is often displaced downstream and spatially separated from the upper-level streamer. The approach for linking RWB to the extreme precipitation seasons in this study does not appear to directly account for this fact.

TECHNICAL CORRECTIONS

line 43: "always" -> "long"

line 46: remove "a high number of"

line 57: "The factors..." Perhaps start a new paragraph here?

line 72: "aggregation" -> "accumulation"

line 81: "The grand" -> "A large"

line 84: "state of the art" -> "scientific understanding of this topic"

line 90: "this chain of events" -> "the chain of events governing precipitation"

line 93–94: I suggest inserting citations immediately after the corresponding phenomenon in the list. For instance, "cyclones (Pfahl and Wernli 2012), fronts (Catto et al. 2012), warm conveyor belts (Pfahl et al. 2014)..."

line 98: would "the frequency and intensity of the precipitation it produces" be more precise than "its frequency and its intensity"?

line 175: "mainly" -> "predominantly"

line 178: "me" -> "be"

line 179: "Indian Ocean)"

line 186: insert "evident in" after "are"

line 193: insert "results for" after "Only"

line 213: Insert "the number of" before "ratio of"

line 222: "few more" -> "a small increase in the number of"

line 232: "the grand" -> "a large"

line 272: "depict" -> "correspond to"

line 273: "includes the track of" -> "corresponds to"

line 352: remove "is" after "It"

line 360: "highlight the important link" -> "suggest links"

line 361–363: delete "Pfahl et al. (2014) showed that" and insert the (Pfahl et al. 2014) at the end of the sentence.

line 377: "make" -> "made"

line 381: insert "necessarily" after "should not"; replace "in the sense that" with "because"; replace "is due" with "can be due"

line 416: "formation" -> "occurrence"

line 439: "the scarcity" -> "climatological infrequency"

line 440: "contributes to" -> "can result in"

line 443: "to moist plumes that originate" -> "transports of moist air"

line 487: "methodology" -> "method"

line 492–493: "considering their…" This is awkwardly worded. Please rewrite.

line 495: insert ", respectively" after "tropics"

line 512: "strongly" -> "highly"

Figure 2: "rainfall" should be changed to "precipitation"

Figure 4: Recommended edit to the caption: "and (b) the ratio of the number"

Figure 5: What is a precipitable day?

Figure 11: The panels should be labeled (a) and (b).

Figure 12: It is unclear to me what you mean by "illustration started from the patch presenting the lowest ratio"

**REFERENCES**

de Vries, A. J., H. G. Ouwersloot, S. B. Feldstein, M. Riemer, A. M. El Kenawy, M. F. McCabe, and J. Lelieveld, 2018: Identification of tropical-extratropical interactions and extreme precipitation events in the Middle East based on potential vorticity and moisture transport. J. Geophys. Res.: Atmos., 123, 861–881, doi:10.1002/2017jd027587.

Martius, O., E. Zenklusen, C. Schwierz, and H. C. Davies, 2006: Episodes of Alpine heavy precipitation with an overlying elongated stratospheric intrusion: A climatology. Int. J. Clim., 26, 1149–1164, doi:10.1002/joc.1295.

Moore, B. J., D. Keyser, and L. F. Bosart, 2019: Linkages between extreme precipitation events in the central and eastern U.S. and Rossby wave breaking. Mon. Wea. Rev. (in press), doi:10.1175/MWR-D-19-0047.1

———————————————

---

## Author Response (AR1)

**Final author comments for wcd-2020-28**

Extreme wet seasons – their definition and relationship with synoptic scale weather systems

E. Flaounas, M. Röthlisberger, M. Boettcher, M. Sprenger and H. Wernli

5   We thank both reviewers for their detailed comments, which are very helpful for further improving our manuscript. In particular, several figures have been revised to increase the clarity of the presentation of the results.

**Reviewer #1**

*This thorough paper introduces a novel method to investigate seasonal precipitation extremes and the large-scale and*
10 *synoptic conditions that give rise to these. This is an important study as such seasonal extremes can be responsible for a number of socioeconomic impacts.*

*The paper is mostly clear and well-written, with a few places that could use some clarification. I give specific comments below.*

We are thankful for the positive and constructive review. We took into account all comments and corrections.

*1. Most of the English usage is British English, but there are some examples of "characterize". Please ensure consistency, as per the WCD instructions.*

Done. English style is now consistent throughout the text.

*2. Line 68: "signalizes" -> "signals".*

25   Done.

*3. Line 87: The use of the word "provoke" is a little strange – perhaps use "give rise to" or "produce".*

Done.

*4. Line 174: I'm not sure what is meant by the "half of the month". This is not referred to elsewhere – is this a mistake?*

Indeed, this is a mistake, "half of the" has been removed to better match the content of the figure.

35   *5. Line 187 and throughout: please use capitals for Northern Hemisphere and Southern Hemisphere.*

Done.

*6. Throughout: The use of the word "amount" for precipitation, might be better as "volume".*

We chose to retain the word "amount" as this term is more commonly found in the scientific literature.

*7. Line 219: The language here is a bit unclear – "contrasting precipitation amount ratios". I'm not sure what this means.*

45   The sentence has been rephrased.

*8. Line 220: But it seems that in parts of the ITCZ, there are high ratios of both the extremes and the wet days. One thing that might be useful in this figure (Figure 4) could be to show the climatological precipitation in fine contours (or just a couple of contours), so that the reader doesn't have to check back to the climatology to see how well things match up.*

Thank you for the suggestion. Figure 4 now includes two isohyet contours: for 500 and 1500 mm (see new version below). This will help the reader to identify areas of frequent and scarce precipitation. In line 220 we removed the word "only" to avoid absolute statements.

[Figure]

**Revised Figure 4 (a)** Ratio of precipitation amount of extreme seasons with respect to the seasonal average, and **(b)** the ratio of the number of daily precipitation extremes included in an extreme season with respect to the seasonal average. **(c)** as **(b)** but for the number of wet days. Dashed and solid contours depict annual average precipitation of 500 and 1500 mm.

60    *9. Line 232: It is stated here that most of the ratios exceed 1, but the figures do not show values below 1. If there are values below 1, the contour intervals on the figure should reflect that and allow the reader to see where this occurs.*

Thank you for this careful comment. The colorbar of Fig. 4 was changed to include values below 1.

65    Only 3% of all grid points have a wet day ratio < 1. The majority of these wet day ratios (75%) range between 0.95 and 1, with a median of 0.98.

For daily precipitation extremes, 13% of all grid points have ratios < 1. The majority of these ratios (75%) range between 0.8 and 1 with a median of 0.98.

70

Ratios < 1 are now mentioned at the end of the paragraph:

*"Before further discussing these patterns, it is noteworthy that 13% of all grid points feature ratios of daily precipitation extremes below 1 (Fig. 5a). These values are concentrated in areas of scarce precipitation and are depicted by grey colors in*
75    *Fig. 4b. For wet days, ratios below 1 are even less common, they occur only for 3% of all grid points and typically exhibit values between 0.9 and 1 (Fig. 5a). In contrast to daily precipitation extremes, these grid points are scattered across areas of frequent precipitation (e.g. ITCZ and storm tracks), where wet day ratios are close to 1, i.e. where extreme seasons occur with roughly the climatological value of wet days."*

80    *10. Figure 5: I really like this way of characterising the extreme seasons. However, it is very difficult to tell the difference between the green/cyan colours. As such some of the writing around this figure is difficult to understand.*
Colours in the figure have been changed (see new Fig. 5 below) and the five points below (a)-(e) have been adequately addressed as suggested.

85    *a. E.g. line 243: The cyan colour referred to over equatorial Africa to me looks like the light blue from the bottom right bin. So it would appear to have a high ratio of daily extremes and low ratio of wet days. Especially as "cyan" is referred to again to describe this same colour on line 253.*

*b. It is mentioned in multiple places about the wet day ratio less than 1.2, but the 9-panel bins show "<2".*
90

*c. Line 259: "20 to 60% more wet days". This is confusing as everywhere else the ratios are referred to – please change this to be similar to previous.*

*d. In this figure it may be useful to also have the climatological precipitation contours.*
95

*e. In panel (a) there is typo in "precipitation".*

[Figure]

**Revised Figure 5 (a)** Probability density function of the number of ratios of daily precipitation extremes and wet days for all extreme seasons and for all grid points (ratios with respect to the seasonal average). Vertical dotted lines correspond to ratios of 1.2, 1.6, 2 and 3. **(b)** Attribution of grid points to nine categories of pairs of ratios of the number of wet days and of daily precipitation extremes. Dashed and solid white contours depict annual average precipitation of 500 and 1500 mm, respectively. Dotted lines in **(a)** show the category boundaries used in **(b)**.

*11. Line 276: El Nino and La Nina -> El Niño and La Niña.*

Done.

*12. Section 4.2: I would be interested to know the sensitivity of the results to defining these core periods. The core periods end up being typically longer than the initially 90 days anyway, so what is the impact of not worrying about it? I suppose the main difference will be when looking at the overlap of the patches with the synoptic systems. It does seem that the method ends up becoming rather complex, so it would be good to know if this extra complexity is necessary.*

Thank you for this insightful comment. The reviewer quite rightly notices a key advantage of defining these core periods: Considering the full duration of the extreme wet periods would make the matching with weather features somewhat fuzzy, as of course numerous weather features would be matched to an extreme season patch despite not occurring during an extremely wet 90-day period. The definition of a core period helps to ensure that the overwhelming majority of weather features that are matched to the extreme season patches, indeed occur during an extremely wet period at the grid points where they occurred. We agree with the reviewer that the definition of a core period adds indeed to the complexity of our methodology, but it assures that patches are temporally and spatially representative of an extreme season. We added the following remark in section 4.2 (see also next comment):

"*It is noteworthy that core periods may not include days with locally intense precipitation events that don't affect a large fraction of the patch area. The intention of the core period is to consider precipitation in the entire larger-scale area of the extreme season patch, and to identify the time period that is most important for precipitation in the patch as a whole.*"

Given that WCD provides an opportunity for online discussions, we would like to further deepen the discussion on the methods that we use. The complexity of our methodology originates from the fact that we adopt a flexible definition of extreme seasons, i.e. using a flexible 90-day period of maximum accumulation of precipitation. In an alternative methodology, we could identify extreme seasons by predefining this time period. For instance, we could identify extremely wet summers (defined as JJA in the Northern Hemisphere) at every grid point. Then, building the patches would be a straightforward procedure and "free-of-complexity", by simply connecting neighbouring grid points that present extreme summers in the same year. The simplicity of this alternative method is intriguing, but it also comes with shortcomings that we overcome with our approach: first, the patch building and analyses would need to be repeated for each season (e.g. for winters, summers, autumns and springs), whereas with our approach, we can obtain a single global view on very wet 90-day periods. Even more importantly, the precipitation season of a specific region might not always fit with the fixed seasons. For instance, the onset of the African monsoon is in the beginning of July. As a result, an extremely wet monsoon season that has a late onset risks of not being documented as an extremely wet season since most of the precipitation will take place in autumn and not in summer. In summary, our method allows high flexibility in the definition of extreme seasons with the price of increasing the complexity. This is a compromise, but it is also where the novelty and strength of our method lies: it provides a new, flexible definition of seasonal precipitation and proposes an approach to define the spatial extent of a wet season. We hope that future studies will build on this, and these studies might further refine the detailed aspects of the patch building and propose alternative metrics to quantify the characteristics of the identified seasons.

*13. In figure 7, because there are fewer grid points contributing, the total precipitation is less, but some of those points may be experiencing their largest precipitation at that time. How is this taken into account?*

This comment addresses directly the motivation of this study. Indeed, a day may be excluded by the core period even if a certain grid point is experiencing high precipitation. Also, in connection with our reply to the previous comment, our motivation here is to obtain patches that are representative for the temporal and spatial scales of seasonal precipitation. Applying additional criteria that force core periods to include local intense precipitation events would bias our method towards singular events and would thus eclipse cases where extreme seasonal precipitation is more due to the aggregation of "moderate events".

*14. Line 249: Why has this particular case been chosen?*

We assume here that the Reviewer refers to line 349. Both cases were chosen for their similarity and for sharing the same characterisation of extreme seasons in Fig. 5. We found that the additional "Portuguese case" would provide some insights into the spatial structures and variability of the patterns that produce extreme seasons in the subtropics/mid-latitudes. Nevertheless, it would be too much to include additional cases in each subsection of section 5. We included the following phrase to justify our choice:

"*Figure 8 provides insight into the complex relationship between precipitation, cyclones and TMEs for this patch, but also for another patch of similar latitudinal extent and orientation that affected the Iberian Peninsula in late autumn 1989 (note that this additional patch is not depicted in Fig. 7).*"

*15. Line 387: Please consider rewording this to make it clear it is a ratio.*

Done.

*16. Case studies: It is a bit confusing that the different cases show different things – it is hard to compare. Why is the Arctic case (Fig. 10) shown as a precipitation anomaly ratio rather than the total precipitation as in the other cases?*

Our choice of case studies meant to cover different latitudes, but also to demonstrate the implications of different weather systems. In this regard, the figures in Section 5 are not consistent, i.e., they present different fields. We agree with the reviewer that this can be confusing for the reader. To lighten section 5, we have removed section 5.4 and thus we now present three cases instead of four. In these three cases, we consistently present the relationship between the patches, the four weather systems and precipitation in terms of ratios and climatological anomalies of occurrence (also according to the query of the second Reviewer). This makes the new Figs. 8, 9 and 10 more consistent to each other. Please find the revised Figs. 8-10 towards the end of this document.

*17. Line 409: "same dates" and grid points.*

Done.

*18. Lines 430-432 (first sentence): This information would be better in the introduction or methods.*

These sentences have been rephrased.

*19. Line 440: Remove "relatively". There are relatively many WCBs even though there are few in absolute terms.*

Done.

*20. Line 443-444: Again, this information could be relocated.*

We agree that these phrases better suit the introduction. However, we prefer to retain these phrases to ease understanding of readers who are not familiar with TMEs to interpret the figures' content.

*21. Line 449: "less" -> "fewer".*

Done.

*22. Figure 12: There are places where the ratio is less than 1 (especially for the RWB), but this is not mentioned. Why might you expect fewer than normal of these systems? Could this vary depending on the time of year?*

Thank you for this insightful comment. There are indeed several patches where the ratio of the weather features is below 1. First, we changed the colorbar to ease contrast between patches with ratios below and above 1 (see revised Fig. 12 below). In addition, we added the following discussion at the end of Section 6:

"*It is noteworthy that in all panels of Fig. 11 there are several patches where the ratios are below 1. This indicates that in the core periods of these patches fewer weather systems occurred than in the climatology. This can plausibly occur if the considered system is not decisive for extreme seasonal precipitation. In such a case, the frequency of occurrence in extreme seasons might be close to the climatological average, i.e. the ratio varies randomly around 1. For instance, the patch covering large part of the great Australian Bight, at the central-south side of Australia, has a ratio below 1 for TMEs (Fig. 11c) whereas cyclones and WCBs have relatively high ratios of 1.6 and 2.2. It is plausible that TMEs do not play a crucial role to the formation of this extreme season compared to other more important contributions from cyclones and WCBs. It is finally noteworthy that we adopted a phenomenological approach to assess the contribution of specific weather systems to the extreme seasons, which only considers the occurrence of a weather system (categorical yes or no) but not specifically its associated precipitation. As a result, it cannot be excluded that a specific weather system might strongly contribute to the formation of an extreme season, even if its seasonal occurrence frequency is lower than in the climatology.*"

[Figure]

**Revised Figure 11** All extreme wet season patches are coloured according to their overlapping frequency ratios with specific weather systems (relative to the climatology). Panels in the right column show the latitudinal distribution of the overlapping ratios, as zonal averages within +/- 7.5° latitude. Patches may overlap between each other; to allow higher visibility for patches with highest ratios, the overlay of the patches in all panels started from the patch with the lowest ratio.

*23. Figure 12 again: Since you have a large number of patches, with central months at different times of the year, I wonder what the figure would look like if this was taken into account? So for the NH midlatitudes, you could make two figures – one for extreme patches with central month in winter, and one for summer. This would surely help to answer the question of whether seasonal extremes are caused by the same mechanisms in different times of the year in the same location. Saying this I realise that I am suggesting making things even more complex despite previously suggested less complexity. However, I think this would be a very interesting addition.*

Thank you for the suggestion. We agree that such a procedure would further complicate the presentation of the results and
235  would slightly escape the purpose of section 6 to provide a compact global overview of the relationship between weather
systems and patches. But we also agree that performing the same analysis separately for DJF, MAM, JJA and SON provides
an interesting perspective and a clearer impression of the monthly distribution of the patches per weather system. We
therefore include these figures in this reply document (see next 4 pages), but we don't include them in the paper.

[Figure]

240  As in revised Fig. 11 in the paper, but here only for patches with a central date in MAM.

[Figure]

As in revised Fig. 11 in the paper, but here only for patches with a central date in JJA.

245

[Figure]

As in revised Fig. 11 in the paper, but here only for patches with a central date in SON.

250

[Figure]

As in revised Fig. 11 in the paper, but here only for patches with a central date in DJF.

255   *24. References: There are quite a few errors in the references, where the "running title" is given as well as the proper title: e.g. Catto et al, Feng et al, Leung et al, as well as other typos.*

We apologize for these mistakes; references are now corrected.

**Reviewer #2**

GENERAL COMMENTS

*This study provides a novel analysis of what the authors refer to as extreme precipitation seasons, defined as 90-day periods during 1979–2018 exhibiting especially large precipitation accumulations. A global climatology of these seasons is constructed and their characteristics are examined through statistical analysis. Contemporaneous global climatologies of warm conveyor belts, tropical moisture exports, breaking Rossby waves, and cyclones are employed to examine dynamical processes that con-tribute to the extreme seasons.*

*Overall, I found this study to be interesting and novel, and I believe that the topic fits within the scope of Weather and Climate Dynamics. The methods developed to identify extreme precipitation seasons and extreme season patches are innovative and novel, though, in my opinion, somewhat complicated. This is the first study to construct a global climatology of extreme precipitation seasons and to attempt to relate them to different types of weather systems. I believe that the study addresses important gaps in scientific understanding regarding the occurrence of extreme precipitation seasons. Despite the strengths of this study, there are a number of issues that need to be addressed with regard to the clarity of the writing, interpretation of the results, the methodology, and the background discussion.*

Thank you for the careful reading of the manuscript, for the positive review and the constructive comments. They were all very helpful to improve the quality of our analysis and presentation of our results.

SPECIFIC COMMENTS

*Abstract: The abstract is quite lengthy and complicated. I recommend simplifying and shortening it.*

The abstract is now shorter and more direct in describing the main results.

*line 43: I recommend being more specific regarding the socioeconomic impacts of these events.*

The sentence has been rephrased to: "… due to their relevance for a variety of socio-economic aspects including damages to infrastructures and loss of life".

*line 47: I suggest also mentioning climatological studies of relationships between PV streamers/breaking waves and precipitation extremes (e.g., Martius et al. 2006; deVries et al. 2018; Moore et al. 2019).*

Done.

*line 51: It is unclear what exactly you mean by 'environmental risks' in this context. Please clarify.*

Sentence has been rephrased to:

"However, socio-economic impacts related to precipitation are not limited to the occurrence of single, outstanding extreme precipitation events, but they are also potentially related to accumulated precipitation on longer timescales."

*line 53: Specify what impacts the hurricanes caused and the coastal regions of the United States that they affected.*

We added the following:

"… *causing damages of the order of 370 billion dollars and loss of human life (Halverson, 2018; Taillie et al., 2020) ...*".

*line 53–54: Note, however, that the season did include several extreme-rain-producing hurricanes.*

310 Thank you for this information. It is now included.

*line 54: Specify what the 'main impact' was? Was it prolonged regional flooding?*

This phrase has been removed.

315

*line 56: Please provide a reference for this statement.*

Done.

320 *line 111: Please explain why this model-based dataset was used. Also, please state any caveats that must be considered when using coarse-resolution model-based precipitation data.*

The following was added:

325 "*Using a model-based instead of an observation-based dataset has the advantage of providing daily fields with continuous spatial coverage over both land and maritime areas. In addition, it assures consistent precipitation fields with atmospheric dynamics. On the other hand, global reanalyses have a rather coarse grid spacing, permitting only the analysis of precipitation related to synoptic-scale weather systems.*"

330 *line 155–158: In my view, the authors have not provided sufficient context and background information to motivate examination of relationships to these different weather systems. This sentence is inadequate in this regard and does not fully and accurately describe the influence that these systems can have on precipitation. For instance, the authors fail to mention that PV streamers and cut-offs have also been found to be linked to strong water vapor transports and dynamical lifting. The four weather system types and their dynamical relationships to precipitation extremes should be described in more detail in*
335 *the introduction section. Also, it could be worthwhile to describe inter-relationships between the four types of systems.*

We agree with the reviewer that the relationship between dynamical processes and precipitation should be further detailed. We changed the paragraph accordingly and added more information:

340 "*All four weather systems are well known to be related to heavy precipitation. Precipitation in the vicinity of these systems is the outcome of a rather complex interaction of dynamical processes that differ between the four systems. For instance, cyclones are known to be responsible for a large part of global precipitation (Hawcroft et al., 2012; Pfahl and Wernli, 2012). The precipitation within cyclones may be attributed to a variety of processes such as deep convection in their center (e.g. in the eyewall of tropical cyclones) and to a combination of convective and stratiform precipitation along the frontal structures*
345 *of extratropical cyclones (Catto and Pfahl, 2013). Especially concerning frontal structures, WCBs can be identified as distinct airstreams that produce high amounts of stratiform and in some cases also convective precipitation (Browning et al., 1973; Flaounas et al., 2017; Oertel et al., 2019). Precipitation due to WCBs affects both the central region of a cyclone and the associated fronts (Catto et al., 2013; Catto and Pfahl, 2013; Pfahl et al., 2014). TMEs foster precipitation indirectly by supplying moisture that may rain out when reaching a region with dynamical or orographic forcing for ascent. Finally, RWB*
350 *can also lead to long-range transport of water vapor, impose large-scale lifting, and reduce static stability in the lower and middle troposphere, favoring thus intense precipitation (Martius et al., 2006; de Vries et al., 2018; de Vries 2020). Sometimes these weather systems occur simultaneously. For instance, RWB may lead to the formation of cyclones that in turn may include WCBs. Therefore, it is an ill-posed problem to determine the separate contribution of these weather systems to*

*total precipitation. However, the objective identification of these weather systems in gridded datasets and counting their seasonal frequency of occurrence may provide interesting insights into their role in extreme wet seasons.*"

*line 168: I suggest using a consistent term for the extreme seasons throughout the paper. Use either "extreme wet season" or "extreme precipitation season" but not both.*

We now use extreme wet seasons throughout the text.

*169–171: While I understand your justification for classifying these seasons as extreme, I am still unsure whether I agree with it. If the seasonal precipitation does not deviate much from climatology, then it really is indicative of an ordinary precipitation season. Are there ways to avoid inclusion of so many secondary seasons in the dataset? Could you use more restrictive criteria to identify secondary extreme seasons? Could you just consider the primary extreme seasons and not the secondary seasons?*

The reviewer correctly identifies a major caveat of our extreme wet season identification scheme: It does not always live up to a proper statistical definition of the word "extreme". However, we deliberately choose to also identify secondary extreme wet seasons because this allows us to perform analyses which, in our opinion, yield more interesting and more relevant results than if we focused solely on primary extreme seasons. In fact, secondary seasons are almost equally important to primary seasons in terms of precipitation amount and therefore also – potentially - in terms of impacts. In addition, if we only focused on primary extreme seasons, this would render the identification of spatially coherent extreme wet season patches, and hence also their matching with weather features, less meaningful, since the patches would then each cover only a very limited number of grid points.

We agree with the Reviewer that in areas where many secondary seasons are identified, these seasons could be considered as "ordinary" (at least from the point of view of the statistical probability of occurrence). However, this also reflects an interesting, central result of our study, i.e. we identified areas where it is "hard" to get distinct periods in terms of precipitation amount. Finally, our Fig. 2a allows the reader to have insights into the likelihood of an area to experience high seasonal precipitation amounts. By identifying e.g. four extreme seasons in our 40-year dataset, in a certain grid point, this suggests that this grid point may experience high seasonal precipitation amounts roughly once every 10 years.

In some sense, by defining extreme wet seasons in this way, we trade some mathematical rigour in their definition for results that we find meaningful and relevant and that could not have been obtained otherwise. We explicitly point to this caveat of our identification scheme in section 2.1:

"*This suggests that in these regions, seasonal precipitation typically varies only by fractions rather than multiples of the climatological mean. Therefore, numerous 90-day periods fall within our definition of secondary "extreme wet seasons". These periods reach almost the same accumulated precipitation as the locally wettest period and, therefore, we choose to use the terminology "extreme wet seasons" also for these periods throughout this manuscript.*"

*line 178: Perhaps insert "and occur most frequently" after "most intense"?*

Done.

*line 184: "This suggests..." I do not see how a lack of a sharp land–sea distinction itself suggests that a given region is influenced by atmospheric rivers and cyclones. It would be more precise to say that the lack of a distinction suggests that a region is influenced by landfalling systems originating over the ocean, such as extratropical cyclones and atmospheric rivers.*

Corrected as suggested.

*line 194–197: Apologies for my confusion, but I am having trouble reconciling this sentence with the previous sentence. If only results for primary seasons are presented, then how can there be multiple extreme seasons at a given grid point.*

405

Thank you for the careful reading and apologies for this typo. Indeed, Fig. 4 does not include any secondary seasons. It is now corrected.

*line 222: "arid areas": I suggest providing specific examples of these areas to aid the reader.*

410

Figure 4 also shows the average annual precipitation. So we added the following: "... occurs rarely (outlined by dashed contours in all panels of Fig. 4)".

*line 225: "climatologically wet regions": I suggest providing specific examples of these regions to aid the reader.*

415

We added the following:

"... *climatologically wet regions (such as in the tropics, within the solid contours of Fig. 4).*"

420 *line 273: Please provide references for the 2010 and 2017 hurricane seasons.*

References to Beven and Blake (2015) and Taillie et al. (2020) were added.

*line 351: It is not clear to me how unusual the frequencies of cyclones, streamers, and TMEs depicted in Figs. 8 and 9 are for*
425 *those regions and seasons. It would be helpful to compare the feature frequencies to the climatological frequencies for the timeperiods, as was done in Fig. 10.*

Thank you for this comment. Also in response to a comment from the first Reviewer, we have revised section 5. To ease the reader, we now present three cases instead of four, excluding the monsoon example (Fig. 11 in the original submission).

430

Figures 8, 9 and 10 have been revised as suggested and the text in section 5 has been adapted to the new figures. Here below follow the new figures showing ratios of seasonal precipitation and frequency anomalies with respect to climatology for the occurrence of weather features. In addition, Figs. 8 and 9 also include examples of case studies of high amounts of daily precipitation.

435

[Figure]

**Revised Figure 8** (a) Ratio of accumulated precipitation during the period 27 November 1992 to 1 March 1993 with respect to climatological values for the same time period (in colour) for an extreme wet season patch affecting the US west coast (dotted area). Red (green) contours show areas with positive anomalies of cyclone (TME) occurrences with respect to climatology. Contours start from 5% and have a 5% of interval. **(b)** as **(a)** but for the period 2 October 1989 to 9 January 1990 for an extreme wet season patch affecting the Iberian Peninsula (dotted area). **(c)** 24-hour accumulation of precipitation from 1200 UTC 28 December 1992 to 1200 UTC 29 December 1992 (in colour). Red contours show sea level pressure at 0000 UTC 29 December 1992 (starting from 1015 hPa and with a step of -3 hPa). Green contours show areas with TMEs and blue contours shows areas with WCB ascent. **(d)** as in **(c)** but at 0000 UTC 26 December 1989.

[Figure]

**Revised Figure 9 (a)** Ratio of accumulated precipitation during the period 27 November 2010 to 3 April 2011 with respect to climatological values for the same time period (in colour) for an extreme wet season patch affecting Australia (dotted area). Red (green) contours show areas with positive anomalies of cyclone (RWB) occurrences with respect to climatology (shown are anomalies of 10 and 20% for cyclones and 20% for RWB). **(b)** 24-hour accumulation of precipitation from 1800 UTC 3 February to 1800 UTC 4 February 2011 (in colour). Red contours show sea level pressure at 1800 UTC 4 February 2011 (starting from 1006 hPa and with steps of -2 hPa). The grey dashed line shows the track of tropical cyclone Yasi, while its position of cyclolysis is represented by the cross symbol.

[Figure]

**23 Jul 2016 - 26 Sep 2016**

**Revised** Figure 10 Ratio of accumulated precipitation during the period 23 July 2016 to 26 September 2016 with respect to climatological values for the same time period (in colour). Red contours show areas with positive anomalies of cyclone occurrences with respect to climatology (shown are contours of 5 and 10%). The spatial extent of the patch is represented by the dotted area.

*line 363–364: "However, the two..." It is unclear to me what the purpose of this sentence is.*

It has been rephrased to: "However, the two exemplary cases in Figs. 8c and 8d also show... "

*line 364–365: "The synergy..." The meaning of this statement is ambiguous to me. Which processes are you referring to?*

It has been rephrased to: "The synergy of cyclones and WCBs is responsible for classifying these periods as extreme wet seasons."

*line 365: "Finally, most..." Mention that this statement applies specifically to the 1992–1993 event.*

This phrase has been removed.

*line 365–367: "However, this comes..." What is the basis for this statement? Please provide a supporting reference.*

This phrase has been removed.

480   *line 373: "In this region..." This is not true. Cyclones can and do occur at these latitudes, as clearly depicted in Fig. 9.*

It has been rephrased to: "... a region where Coriolis forces are too weak to favour cyclogenesis."

*line 373–374: "However, RWB..." A figure reference is needed in this sentence.*

485   Done.

*line 374: By "upper-tropospheric systems" do you mean elongated PV streamers associated with RWB? If so, consider saying "The upper-level PV streamers resulting from the events". Upper-level is more accurate than upper-tropospheric here given*
490   *that these systems are defined as narrow filaments of stratospheric high-PV air.*

Changed as suggested.

*line 389: The anomalous warmth could also reflect frequent poleward excursions of warm, moist air into the Arctic that*
495   *supported the precipitation within the patch.*

This is a very interesting suggestion. However, failing to find past studies to support this statement we chose not to add it in the paper.

500   *line 392: "probably reflect..." this assertion does not appear to be supported by any evidence.*

The phrase is changed to:

"*Evidently, such conditions can lead to extreme wet seasons in the eastern Arctic and are similar to the ones leading to a*
505   *rainier future regime in the Arctic region (Bintanja, 2017).*"

*line 416: What do you mean by "the largest part of the world"?*

This phrase has been changed to:
510
"*Most latitudes except in the tropics...*"

*line 417: Does this imply that the cyclone climatology used in this study also includes tropical cyclones and other tropical low pressure systems in addition to extratropical cyclones? Is there any distinction made in the climatology between*
515   *extratropical and tropical systems?*

We make no distinction between tropical, subtropical or extratropical cyclones. We included the following:

"*The origin of these maxima cannot be attributed clearly to either tropical, subtropical or extratropical cyclones.*
520   *Nevertheless, Fig. 3 shows that these regions experience their extreme seasons in the colder months of the year and thus it is rather unlikely that tropical cyclones may contribute to their formation.*"

*line 422–423: I find this sentence confusing. Which result is in accordance with Pfahl and Wernli (2012)? Also, it is a sentence fragment.*
525
The phrase has been changed to:

*"This result suggests that cyclones occurring equatorward of the climatological storm tracks are a key ingredient for extreme wet seasons since they trigger anomalously frequent precipitation extremes in these regions (see also Pfahl and Wernli 2012)."*

530

*line 430: it would be more dynamically accurate to say "baroclinic zones associated with cyclones" instead of "cyclones' frontal surfaces"*

535  The phrase has been deleted.

*line 437: "physical characteristics" is vague. Please specify the physical characteristics that are relevant in this context.*

"Physical characteristics" has been deleted.

540

*line 443–444: "Therefore, TMEs..." This statement strikes me as erroneous. Can you cite a study that supports this claim? My understanding is that a TME will only support heavy precipitation where it encounters a region of strong ascending motion; thus, TMEs should not be expected to produce high amounts of precipitation whenever they reach higher latitudes but rather only under certain circumstances.*

545

Indeed, we meant that TMEs do not trigger convection, but rather favour the production of higher amounts of precipitation. The phrase has been changed to:

*"Therefore, TMEs are expected to favour higher amounts of precipitation whenever they reach areas of strong ascending*
550  *motion in higher latitudes.'*

*line 448–449: "Occasionally, TMEs..." I find this sentence somewhat confusing. Please rephrase more clearly.*

Phrase has been changed to:

555

*"Occasionally, TMEs contribute to the formation of extreme seasons in the Arctic, but the high ratios in this region (Fig. 12c) result from few events during the extreme seasons and even fewer in the climatology."*

*line 454–455: I do not entirely follow this reasoning. The ratios shown in Fig. 12 do not necessarily indicate the strength of*
560  *the contribution of a given type of weather system. They only indicate the degree to which weather system frequencies deviate from climatology during extreme precipitation seasons. It seems to me that it is still possible for systems to produce large portions of the precipitation during extreme seasons even if their frequencies do not deviate substantially from climatology.*

We agree that ratio of occurrence is not necessarily correlated with the precipitation amount. To avoid confusion, we removed
565  the previous sentence so that lines 454-455 do not come as a natural continuation of the physical relationship between RWB and the production of precipitation. The removed lines have been shifted to section 2.3.

*line 457: It would be more precise to say "PV streamers" rather than "filaments"*

570  Done.

*line 459: What do you mean by "RWB into the tropics"? Perhaps it would be more accurate to say "extension of PV streamers into the tropics".*

575  Done.

*line 461–463: "It is noteworthy that..." I really do not understand this sentence. Please clarify.*

It has been changed to:

*"Nevertheless, other weather systems or conditions than RWBs, cyclones and WCBs might be also involved in forming daily precipitation extremes in the tropics and thus be responsible for the formation of extreme seasons (e.g. a very strong ITCZ or warmer sea surface temperatures)."*

*line 463–464: "Finally, the..." This sentence does not make sense to me.*

Phrase has been changed to:

*"Finally, in polar latitudes there are relatively high frequency ratios of RWB that may be directly related to the high frequency ratios of WCBs in same areas (Fig. 12b), especially in the Southern Hemisphere."*

*line 464–466: "Indeed, WCBs..." I do not understand how this sentence connects with the preceding discussion in this paragraph.*

Changing the previous phrase and simplifying this one, we believe that now the connection is clearer.

*line 488: It seems to me, based on the results in Figs. 8–11, that large patches can also result from synoptic-scale weather systems, such as extratropical cyclones and RWB. This should also be mentioned here.*

Done.

*line 499–500: The streamers that form in connection with wave breaking tend to be part of baroclinic waves that are tilted with height. Thus, widespread heavy precipitation produced in association with wave breaking is often displaced downstream and spatially separated from the upper-level streamer. The approach for linking RWB to the extreme precipitation seasons in this study does not appear to directly account for this fact.*

Thank you for this comment. In this study we restrict ourselves to simply quantifying how often weather systems co-occur with the patches during extreme wet seasons, relative to their climatological occurrence in the respective regions. We agree with the reviewer that it is a challenge to quantify the exact amount of precipitation caused by weather systems, especially RWB. By diagnosing the co-occurrence of weather features and patches of extreme wet seasons we expect to consider a large part of precipitation in our patches that may extend beyond the grid points that define RWB features. The causation in our arguments is based on the many previous studies that show in much more detail how the four weather systems are related to precipitation, as now explained in section 2.3 (in response to your comment on lines 155-158 of the original submission).

We are now more explicit on how we calculate the co-occurrence ratios, including the following at the end of section 2.3:

*"A common framework has been applied to quantify the co-occurrence of these weather systems and extreme season patches. This co-occurrence is defined for each patch as the number of grid points of the patch that overlap with a specific weather system (note that all our weather systems are defined as two-dimensional objects), averaged during the core period (see Section 4.2) of the patch. We then show ratios of this co-occurrence during the core period of the considered extreme season (e.g., from 10 Feb to 22 May 1993) with respect to the climatological co-occurrence (40-year average for periods from 10 Feb to 22 May). A more detailed method to quantify co-occurrence would require a direct attribution of precipitation to each weather feature, as done, e.g., by Moore et al. (2019) and de Vries (2020). Nevertheless, this would increase the complexity,*

*since several weather systems may interact to synergistically produce high precipitation amounts, as explained above. Our method thus simply quantifies the co-occurrence of weather systems and extreme wet seasons in the regions identified as wet season patches. Nevertheless, due to the direct relevance of the four weather systems for precipitation, our approach provides insight into the role of weather systems in forming extreme seasons."*

TECHNICAL CORRECTIONS

line 43: "always" -> "long"

Done.

line 46: remove "a high number of"

Done.

line 57: "The factors..." Perhaps start a new paragraph here?

Done.

line 72: "aggregation" -> "accumulation"

Done.

line 81: "The grand" -> "A large"

Done.

line 84: "state of the art" -> "scientific understanding of this topic"

Done.

line 90: "this chain of events" -> "the chain of events governing precipitation"

Done.

line 93–94: I suggest inserting citations immediately after the corresponding phenomenon in the list. For instance, "cyclones (Pfahl and Wernli 2012), fronts (Catto et al. 2012), warm conveyor belts (Pfahl et al. 2014)..."

Done.

line 98: would "the frequency and intensity of the precipitation it produces" be more precise than "its frequency and its intensity"?

Done.

line 175: "mainly" -> "predominantly"

Done.

line 178: "me" -> "be"

Done.

line 179: "Indian Ocean)"

Done.

line 186: insert "evident in" after "are"

Done.

line 193: insert "results for" after "Only"

Done.

line 213: Insert "the number of" before "ratio of"

Done.

line 222: "few more" -> "a small increase in the number of"

Done.

line 232: "the grand" -> "a large"

Done.

line 272: "depict" -> "correspond to"

Done.

line 273: "includes the track of" -> "corresponds to"

Done.

line 352: remove "is" after "It"

Done.

line 360: "highlight the important link" -> "suggest links"

Done.

line 361–363: delete "Pfahl et al. (2014) showed that" and insert the (Pfahl et al. 2014)at the end of the sentence.

Done.

720 line 377: "make" -> "made"

Done.

725 line 381: insert "necessarily" after "should not"; replace "in the sense that" with "because"; replace "is due" with "can be due"

Done.

730 line 416: "formation" -> "occurrence"

Done.

line 439: "the scarcity" -> "climatological infrequency"

735

Done.

line 440: "contributes to" -> "can result in"

740 Done.

line 443: "to moist plumes that originate" -> "transports of moist air"

Done.

745

line 487: "methodology" -> "method"

Done.

750 line 492–493: "considering their..." This is awkwardly worded. Please rewrite.

Sentences were rephrased to:

*"Four weather systems, known to be related to (extreme) daily precipitation events, were used to understand the role of*
755 *synoptic-scale dynamics in forming extreme wet seasons. These systems were objectively identified in the 40-year dataset in*
*order to quantify their overlap with the extreme wet season patches."*

line 495: insert ", respectively" after "tropics"

760 Done.

line 512: "strongly" -> "highly"

Done.

765

Figure 2: "rainfall" should be changed to "precipitation"

Done.

770 Figure 4: Recommended edit to the caption: "and (b) the ratio of the number"

Changed as suggested.

Figure 5: What is a precipitable day?

775

Changed to "wet days".

Figure 11: The panels should be labelled (a) and (b).

780 Done.

Figure 12: It is unclear to me what you mean by "illustration started from the patch presenting the lowest ratio"

The last sentence of the caption was changed to:

785

*"Patches may overlap between each other; to allow higher visibility for patches with highest ratios, the overlay of the patches in all panels started from the patch with the lowest ratio."*

[revised manuscript text omitted]

---

## Author Response (AR2)

Comments from Reviewer 2

GENERAL COMMENTS:

*The authors have satisfactorily addressed my comments on the previous version of this manuscript. In my opinion, the manuscript has been improved substantially and is acceptable for publication in Weather and Climate Dynamics. I have just a few minor comments for the authors to consider (see below). I look forward to the publication of this nice paper.*

We would like to thank the Reviewer for all his/her help to improve the paper.

SPECIFIC COMMENTS:

*line 74-75: It could be helpful to also explicitly state here that the stormy winter of 2013/14 involved serial clustering of extratropical cyclones (e.g., Priestley et al. 2017; https://doi.org/10.1002/wea.3025). Cyclone clustering seems like a highly relevant process for extreme seasons that occur in regions located along/near midlatitude storm tracks. Thus, perhaps this topic deserves a bit more discussion in this manuscript.*

Thank you for suggesting this study. We now reference the study of Priestley et al. (2017) in lines 74-75 and later in section 4.1.

*line 161: Change to "distinct ascending airstreams extending through a cyclone warm sector"*

Done.

*line 196: Insert "midlatitude" before "continental"?*

Done.

*line 430: The plots in Fig. 11 provide a nice global view of the relationships between extreme seasons and the different weather system types. That being said, the maps are rather noisy. Perhaps this is a naive question, but would plotting the average ratio value for all patches at each grid point, rather than overlaying all patches on the map, provide a clearer and cleaner depiction?*

Thank you for this comment. We acknowledge a certain noisiness in Fig. 11 but unfortunately the proposed method did not provide a clearer illustration either. While in certain areas the field of ratios became smoother, other areas became even more noisy. This is due to overlapping patches of contrasting ratios that ended up with a sharp transition of colours. In any case, we agree with the Reviewer that Fig. 11 serves well the purpose of providing a global view of our results and we trust that, despite the certain degree of noisiness, the figure conveys a clear message. Therefore, we choose to keep the figure as it is.

TECHNICAL CORRECTIONS:

*line 89: wave -> waves*

Done.

*line 200: change the semicolon to a comma*

Done.

*line 399: The contours for RWB occurrences in Fig. 9a appear yellow/brown instead of green.*

Thanks, this was a typo, "green" is now changed to "brown".